# Multi-dimensional concept discovery (MCD): A unifying framework with completeness guarantees

**Johanna Vielhaben** *johanna.vielhaben@hhi.fraunhofer.de*
*Explainable Artificial Intelligence Group*
*Fraunhofer Heinrich-Hertz-Institute*

**Stefan Blüecher** *bluecher@tu-berlin.de*
*Machine Learning Group*
*TU Berlin*

**Nils Strodthoff** *nils.strodthoff@uol.de*
*Division AI4Health*
*Oldenburg University*

**Reviewed on OpenReview:** *https://openreview.net/forum?id=KxBQPz7HKh*

## Abstract

The completeness axiom renders the explanation of a post-hoc eXplainable AI (XAI) method only locally faithful to the model, i.e. for a single decision. For the trustworthy application of XAI, in particular for high-stake decisions, a more global model understanding is required. To this end, concept-based methods have been proposed, which are however not guaranteed to be bound to the actual model reasoning. To circumvent this problem, we propose Multi-dimensional Concept Discovery (MCD) as an extension of previous approaches that fulfills a completeness relation on the level of concepts. Our method starts from general linear subspaces as concepts and does neither require reinforcing concept interpretability nor re-training of model parts. We propose sparse subspace clustering to discover improved concepts and fully leverage the potential of multi-dimensional subspaces. MCD offers two complementary analysis tools for concepts in input space: (1) concept activation maps, that show where a concept is expressed within a sample, allowing for concept characterization through prototypical samples, and (2) concept relevance heatmaps, that decompose the model decision into concept contributions. Both tools together enable a detailed global understanding of the model reasoning, which is guaranteed to relate to the model via a completeness relation. Thus, MCD paves the way towards more trustworthy concept-based XAI. We empirically demonstrate the superiority of MCD against more constrained concept definitions.

## 1 Introduction

Explainable AI (XAI) allows to peek insight the black box of inherently complex deep learning models. *Local* interpretability methods are particular valuable, as they measure attributions for an individual instance, which are easily comprehensible for any kind of end-users, see (Covert et al., 2021; Lundberg & Lee, 2017; Montavon et al., 2018; Samek et al., 2021) for reviews. For example, local methods make a prediction interpretable on the level of single images or individual bank customers for an image or credit risk classifier, respectively. Importantly, the commonly employed *completeness axiom* (attributions sum up to the model prediction) ensures a meaningful interpretation of attributions (Lundberg & Lee, 2017; Sundararajan et al., 2017). However, to actually comprehend the model reasoning we require a *global* model understanding, which reliably explains the model behavior across multiple instances (e.g. a group of female vs. male bank customers). We stress that it is not viable to require an end-user to aggregate local attributions into common

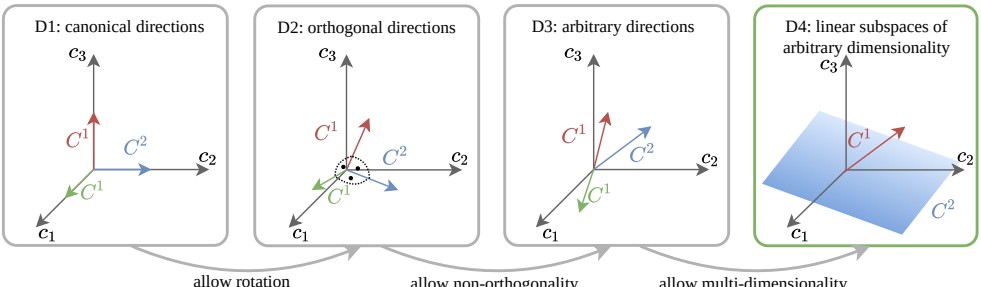

Figure 1: We strive for the most general decomposition of the hidden feature space, spanned by the neurons $c_1, c_2, c_3$, into linear structures that form the concepts $C^i$. The most constrained approach is to identify concepts with single neurons (D1), i.e. directions in feature space aligned with canonical basis vectors. If one allows for arbitrary rotations of the concept directions, one arrives at D2. Leaving aside the orthogonality constraint, D3 allows concepts to form arbitrary directions in feature space. Finally, allowing concepts to form multi-dimensional subspaces, we arrive at the most general approach D4. Previous concept-based methods are based on D1, D2 and D3. We choose the most general approach D4, to discover concepts that are *faithful*.

model features (concepts). Such a procedure is prone to human confirmation bias and it is not clear how the imagined concepts align with the actual model reasoning. This urges for novel *local* and *concept-based* interpretability methods, which allow to understand shared model structures (used across multiple samples) for an individual instance. This idea was first formalized by Kim et al. (2018) and further developed by ACE (Ghorbani et al., 2019) and its successors (Yeh et al., 2020; Zhang et al., 2021). Crucially, our work re-introduces *completeness* within the context of concept-based explanations. Thereby, concepts obtained within our multi-dimensional concept discovery (MCD) scheme are *locally* and *globally* interpretable in terms of a well-defined completeness decomposition. We outline the benefits of MCD in the following paragraphs.

**Concepts as multi-dimensional subspaces** Indisputably, concept discovery in neural networks is inherently linked to structures in the activations of intermediate feature layers. In Figure 1, we illustrate different approaches to decompose the hidden feature space (union of all possible activations) into meaningful concepts, which are mathematically formalized as linear structures. As an illustrative example, we consider the activations of the last convolutional layer just before average pooling and a linear classification head. In this case, the model part remaining after the intermediate feature layer can only exploit linearly separable concepts, hence justifying the linearity assumption here. The most constrained definition (left most panel, D1) is to directly identify concepts with canonical basis vectors of the feature space (Bau et al., 2017). In our example, D1 identifies each convolutional channel with a concept. A slightly more general definition is to allow concepts to lie on orthogonal directions other than the unit axes in feature space (D2). In our example, this means, concepts are formed by orthogonal linear combinations of convolutional channels. Such a concept decomposition can be obtained via a principal component analysis (PCA) of the feature space (Zhang et al., 2021). Going one step further, we disregard the orthogonality constraint and allow arbitrary directions in feature space (D3) (Ghorbani et al., 2019; Kim et al., 2018; Yeh et al., 2020; Zhang et al., 2021). Thereby, we can characterize related concepts which are linearly independent but not orthogonal. This is sensible because in general, the model has no mechanism that enforces orthogonality of concepts (for example different parts of an animal). Allowing for arbitrary multi-dimensional subspaces unfolds the most general definition of a linear decomposition (D4). Coming back to the CNN example, D4 allows a concept

to lie on a hyperplane spanned by multiple directions of the convolutional channels. We argue, that this general approach enables the most *faithful* concepts among D1-D4, as it allows to capture any meaningful linear structure within the hidden feature layer (*benefit 1*).

**Multi-dimensionality ensures concise explanations** Concepts strive to organize the information about the global model reasoning in a concise manner. To this end, we want to cover the relevant feature space with only a few concepts and avoid fragmentation into a large number of low/one-dimensional subspaces. As a first step, we propose a concept completeness score, which measures the fraction of model prediction jointly covered by all concepts. We find that it requires significantly fewer multi-dimensional MCD concepts to reach a specified level of completeness as compared to more constrained concept definitions (D1-D3), i.e., MCD provides more concise explanations (*benefit 2*).

**Re-establishing completeness for concepts** To define concept relevances, in Section 2.3, we uniquely decompose the hidden activations in conjunction with the model prediction into concept parts. To this end, we restrict to a high-level feature layer which is only succeeded by linear operations (e.g. a linear classification head with global pooling). Then, the concept relevances follow a completeness relation (*benefit 3*), i.e., summing all concept relevances equals the final prediction. Thus, we restore the often-desired completeness property mentioned above for concept explanations. We stress, that our concept relevances follow directly from the decomposition into concept parts and do not invoke any additional XAI method nor retraining model parts. Phrased differently, MCD can completely capture the model reasoning solely in terms of linear operations on concept subspaces.

In summary, MCD is a consistent framework to discover *faithful* concepts, which are guaranteed to rely on the actual model reasoning via the completeness relation. Our framework provides several possibilities to investigate the discovered reasoning structure in input space. Thus, we see the main utility of MCD in the domain of model understanding and certification. Concepts provide insights into model behavior that generalize across samples and are therefore a valuable tool for systematic investigations of spurious correlations (model biases) (Lapuschkin et al., 2019; Palatnik de Sousa et al., 2021; Weber et al., 2021), as well as for scientific discovery (Blücher et al., 2020; Hägele et al., 2020; McGrath et al., 2022; Šarčević et al., 2022), where the model serves as a proxy for the unknown relationships in the data.

## 2 Multi-dimensional Concept Discovery (MCD)

We organize this methodological section into three parts: First, we introduce our novel concept definition in Section 2.1. Second, we describe practical concept discovery procedures that align with this definition in Section 2.2. Third, we introduce a concept decomposition and discuss how to construct local and global concept importance that fulfill a concept completeness relation in Section 2.3.

Fig. 2 presents a schematic summary of our MCD framework, which we shortly summarize here. During the training phase, the approach discovers multi-dimensional subspaces (concepts) in the hidden feature space of model, which we mathematically capture as concept bases. These concept bases are obtained from (i) clustering feature vectors (using some particular algorithm) and (ii) characterizing clusters by their dominant principal directions (lower left panel). During the testing phase, we can compare new feature vectors (e.g. from a new test sample) with these concept bases and thereby analyze/characterize all concepts (right panel). Here, we provide two complementary tools: *concept activation maps*, which highlight strongly expressed regions of the concept in input space and *concept relevance maps*, which show how indicative a particular concept is for the class predictions.

### 2.1 Concept definition

Concepts are inherently tied to the hidden representations of intermediate feature layers. For our concept definition, we split the model $f$ into two parts, $f = g \circ h$, where $h$ is the mapping to a hidden feature layer, which is mapped to the prediction by $g$. Our definition then relies on hidden representations $h(\alpha) \in \mathbb{R}^{H \times W \times F}$ of input samples $\alpha$ (height $H$, width $W$ and number of features $F$, see upper left panel in Figure 2).

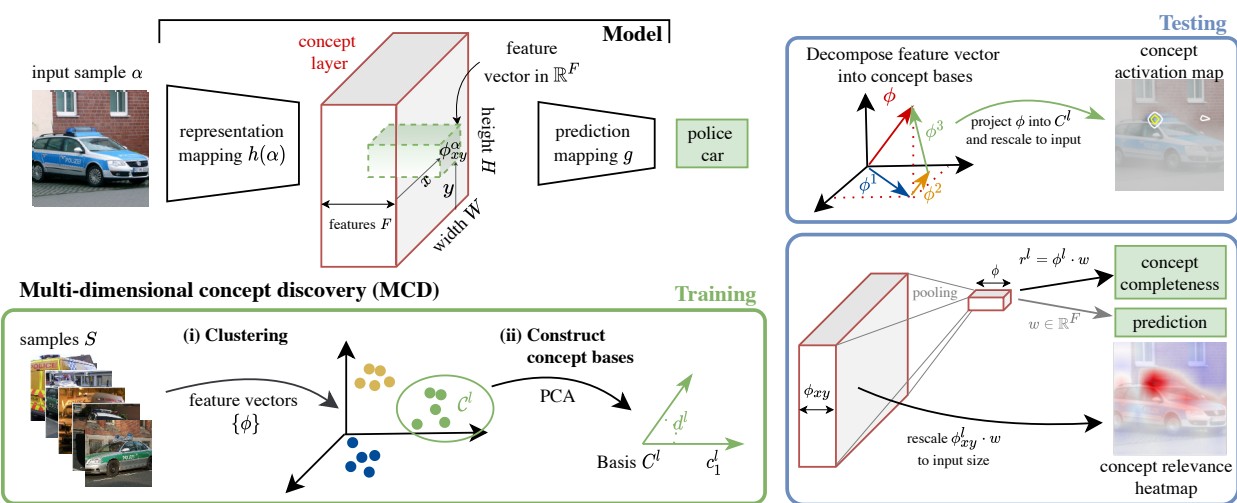

Figure 2: Schematic illustration of the MCD framework for concept discovery. The **upper left panel** illustrates how the model is split into a representation and prediction mapping. Feature vectors are extracted from the representation mapping of a sample. The **lower left panel** illustrates the concept discovery methodology of MCD (Section 2.2). First, randomly choose and cluster a set of feature vectors $\{\boldsymbol{\phi}\}$ from a selection of samples (using any clustering algorithm). Second, construct subspace bases for all clusters $C^l$ via PCA (intrinsic dimension $d^l$). The **upper right panel** corresponds to the construction of *concept activation maps* and the **lower right panel** shows the construction of *concept relevance heatmaps*, both laid out in Section 2.3.

We spatially deconstruct the feature maps $h(\alpha)$ and obtain a feature vector[1] $\boldsymbol{\phi}^\alpha_{xy} \in \mathbb{R}^F$ for each location $(x, y) \in \{1, \ldots, H\} \times \{1, \ldots, W\}$. We now strive to identify concepts as (linear) structures in this $F$-dimensional feature space and pose no additional restrictions (one-dimensionality and/or orthogonality) on the structure of the subspaces.

**Definition 1.** *We define a concept $C^l$ as a $d^l$ dimensional linear subspace in the $F$-dimensional feature space, spanned by the basis vectors $\boldsymbol{c}^l_j$,*

$$C^l = span\left(\{\boldsymbol{c}^l_j | j = 1, \ldots, d^l\}\right) . \tag{1}$$

In particular, the dimensionality $d^l$ can vary among the concepts $l = 1, \ldots, n_c$. We denote the number of concepts as $n_c$ and present a constructive way to determine it in Section 2.3. assume that the concept subspaces are pairwise disjoint.

In Figure 1, we illustrate the linear structures that concepts could possibly form in hidden feature space: from single directions (D1-D3) to linear subspaces (D4). The MCD concept definition above corresponds to D4, which is the most general linear structure, i.e., arbitrarily orientated multi-dimensional linear subspaces. Note, that exploring even more general, non-linear concept structures, such as sub-manifolds in feature space, is an interesting idea for layers where non-linear operations follow. For these, linear multi-dimensional subspaces represent an improvement over the previous, more constrained linear concept definitions in terms of faithfulness. However, for a last hidden layer that is only followed by linear classification head, which we specialize to in Section 2.3, linear subspaces are the most general structure that can be separated, and thus form a faithful model concept definition.

---

[1]Vectors are denoted lower-case bold ($\boldsymbol{\phi} \in \mathbb{R}^F$).

### 2.2 Concept Discovery

Typically, concept discovery, i.e., obtaining concepts as defined by Equation (1), can be subdivided into two steps: (i) cluster a user-defined set of feature vectors $\{\boldsymbol{\phi}_{x,y}^{\alpha}\}$ into clusters $\mathcal{C}^1, \ldots, \mathcal{C}^{n_c}$ and (ii) identify a representative basis $C^l = \{\boldsymbol{c}_j^l | j = 1 \ldots d_l\}$ for each concept cluster $\mathcal{C}^l$ (lower left panel in Figure 2).

**(i) Clustering feature vectors** In principle, any clustering method can be considered to discover concept clusters in feature space. This includes well-established baselines such as k-means clustering or PCA. Both have previously been proposed in (Zhang et al., 2021) to identify one-dimensional subspaces. However, k-means does not incorporate any information about the final objective to identify linear subspaces as opposed to general clusters and PCA is restricted to orthogonal, one-dimensional subspaces. We, therefore, propose a dedicated approach for this particular purpose to discover multi-dimensional linear subspaces and draw on the rich body of literature on *sparse subspace clustering* (SSC) (You et al., 2016a; Soltanolkotabi & Candes, 2012; You et al., 2016b; Elhamifar & Vidal, 2013). SSC clusters datapoints that lie on a union of separate low-dimensional subspaces embedded in a high-dimensional space. It is based on the idea, that a feature vector $\boldsymbol{\phi}_i$ can be expressed as a linear combination of other feature vectors from the same cluster/subspace $\mathcal{C}^l$. As nicely laid out in (Elhamifar & Vidal, 2013), SSC is ideally suited to identify clusters of linear subspaces and provides a number of advantages over standard clustering algorithms, which are directly applied to the data: SSC does not rely on the spatial proximity of the data, it can be implemented robustly against noise and outliers and does not require specifying the cluster dimensionalities in advance.

We start out with a user-specified set of samples $S$ for which we aim to discover concepts. The sample selection $S$ is unrestricted: the user can decide on class-specific samples/concepts or use all training samples to obtain completely class-unspecific concepts. Then, the first step of SSC is to find a sparse self-representation matrix that expresses each $\boldsymbol{\phi}_i$ in terms of a minimal number of other feature vectors $\{\boldsymbol{\phi}_{x,y}^{\alpha}\}$. Second, spectral clustering is applied to the self-representation to obtain clusters $\mathcal{C}^l$. We provide technical details on the particular subspace algorithm in Appendix A.

**(ii) Constructing concept bases** We have now identified clusters $\mathcal{C}^1, \ldots, \mathcal{C}^{n_c}$, which contain all feature vectors $\boldsymbol{\phi}_{x,y}^{\alpha}$ from the training set. Next, we want to obtain general concepts and become independent from the original specific cluster members. To this end, we aim to identify a basis $C^l$ that robustly covers the cluster $\mathcal{C}^l$. By construction, all $\boldsymbol{\phi}_i \in \mathcal{C}^l$ lie within a linear subspace, hence a linear tool like principal component analysis (PCA) constructs an accurate basis $C^l$ for the cluster $\mathcal{C}^l$. We determine the intrinsic dimension $d^l$ of the subspace using a heuristic proposed by Fukunaga & Olsen (1971) and implemented by Bac et al. (2021). The PCA components up to the intrinsic dimension $d^l$ then serve as basis vectors $\boldsymbol{c}_j^l$ for the subspace $C^l$. Given two subspaces, we can quantify their relation in terms of their Grassmann distance, see Appendix B.

### 2.3 Concept decomposition

Previously, we have laid out how to discover an expressive set of concepts $C^l$. Next, we discuss how new features vectors $\{\boldsymbol{\phi}_{x,y}^{\beta}\}$ (obtained from a test set sample $\beta$) and the weights of the final linear classifier layer can be analyzed via a decomposition into concept contributions. To this end, we propose *concept activation maps*, *concept relevance heatmaps* and a *global concept relevance score*. These are complementary tools that form the final concept explanation, informing about the overall meaning of a concept (*concept activation maps*) and its impact on the classification of a specific sample (*concept relevance heatmaps*) or on a global level (*global concept relevance score*).

To ensure that the union of all concepts spans the entire feature space, we define $C^{\perp}$ to be the orthogonal complement of the subspace spanned by all concepts, i.e., $C^{\perp} = \text{span}(C^1, \ldots, C^{n_c})^{\perp}$. To simplify the notation, we identify $C^{n_c+1} \equiv C^{\perp}$. In the following, we assume that the concept subspaces are pairwise disjoint. [2]

---

[2]This assumption was never violated in our experiments. If necessary, this could be enforced by removing the intersection between the subspaces from both and considering it as a separate concept.

**Concept activation maps** quantify the activation of a chosen concept at a certain spatial location in the input space of a sample $\beta$. For this purpose, we decompose the feature vectors $\{\boldsymbol{\phi}_{x,y}^{\beta}\}$ into its unique concept contributions. Since the union of all concepts (including the orthogonal complement) forms a basis for the entire feature space, we can uniquely decompose any feature vector $\boldsymbol{\phi}$ as

$$\boldsymbol{\phi} = \sum_{l=1}^{n_c+1} \sum_{i=1}^{d^l} \varphi_i^l \boldsymbol{c}_i^l \equiv \sum_{l=1}^{n_c+1} \boldsymbol{\phi}^l \,, \tag{2}$$

where $\varphi_i^l$ are the components of $\boldsymbol{\phi}$ in the given basis. Now, one can interpret $|\boldsymbol{\phi}^l|_2$ as a measure for the extent to which a certain concept is expressed in the given feature vector. Performing this step for every feature vector within a sample $\beta$, i.e., $\boldsymbol{\phi}_{x,y}^{\beta} = \sum_{l=1}^{n_c+1} \boldsymbol{\phi}_{x,y}^{\beta,l}$, leads to a *concept activation map* $|\boldsymbol{\phi}_{x,y}^{\beta,l}|_2$ whose spatial dimensions match those of the feature layer. For a fixed sample $\beta$, we normalize $\boldsymbol{\phi}$ such that the maximum length across all elements of the feature layer is 1, i.e., we divide the vectors elementwise by $\max_{x,y}|\boldsymbol{\phi}_{x,y}^{\beta}|_2$. For CNNs, we follow the example of Selvaraju et al. (2020) and compute the corresponding concept activation map in input space by bilinear upsampling in the spatial dimensions. Our concept activation maps extend the concept visualization of Zhang et al. (2021) to multi-dimensional concepts (upper right panel in Figure 2). For the final explanation, we also use them to characterize a concept in terms of prototypical examples. To this end, for each concept $l$, we sort test set samples by the maximum activation $\max_{xy}|\boldsymbol{\phi}_{x,y}^{\beta,l}|$ and choose the top-$k$ samples as *concept prototypes*.

We stress, that our methodology is applicable beyond CNNs. In particular, one can decompose feature representations of any model based on MCD. However, the prerequisite for showing concept (activation) maps in input space is the locality of the trained model, i.e., the ability to associate locations in feature and input space. Whereas this locality is built in as an inductive bias into convolutional architectures, it also emerges for vision transformer models during training, as manifested for example in localized attention maps (Caron et al., 2021). To substantiate these claims, we show the first concept-based explanations for a vision transformer model in Section 4. As a final remark, we emphasize that concept activation maps can be evaluated for any model, including self-supervised pretrained models before finetuning with a classifiaction head, and any layer in the model. They reveal the learned structures in feature space and are only constrained by the restriction to linear subspaces instead of more general non-linear structures.

**Concept relevance heatmaps and completeness relation** As a general requirement, any concept-based XAI method should quantify the *relevance* of a concept in terms of its impact on the classification decision. To this end, we specialize to the last hidden layer, which is only followed by linear operations (e.g., mean pooling and a linear classification head). We discuss the broad class of models to which this applies in the last paragraph of this section and empirically in Section 4.

For a given class, the weight vector $\boldsymbol{w} \in \mathbb{R}^F$ linearly connects the final $F$-dimensional feature space with the scalar class prediction. First, we consider the feature vector after pooling $\boldsymbol{\phi} \equiv \boldsymbol{\phi}^{\beta} = \frac{1}{WH} \sum_{x,y} \boldsymbol{\phi}_{xy}^{\beta}$ ($\boldsymbol{\phi}^{\beta} \in \mathbb{R}^F$) in this very layer (see Figure 2 lower right panel). Now, we are interested in a *local* (per-sample) concept relevance. For this, we can decompose the class logit under consideration, $\boldsymbol{\phi} \cdot \boldsymbol{w} + b$, up to the bias term $b$, as

$$\boldsymbol{\phi} \cdot \boldsymbol{w} = \sum_{l=1}^{n_c+1} \boldsymbol{\phi}^l \cdot \boldsymbol{w} \equiv \sum_{l=1}^{n_c+1} r^l \,. \tag{3}$$

The decomposition above defines a *local concept relevance* $r^l = \boldsymbol{\phi}^l \cdot \boldsymbol{w}$. Aggregating relevances $r^l$ from all concepts recovers the class logit prediction (up to the bias term), and thus, Equation (3) defines a *completeness relation.* [3] [4]

Second, we apply Equation (3) to the feature vectors $\boldsymbol{\phi}_{xy}^{\beta}$ before pooling. This leads to a relevance heatmap $r_{xy}^l = \boldsymbol{\phi}_{xy}^{\beta,l} \cdot \boldsymbol{w}$ that has the same spatial dimension as the feature layer. Importantly, $r_{xy}^l$ reduces to $r^l$ after

---

[3] In the special case of one-dimensional concepts, $r^l$ reduces to the local concept relevance in (Zhang et al., 2021).

[4] We briefly comment on the remaining commonly desired Shapley axioms Lundberg & Lee (2017). The local concept relevance trivially fulfills them since it is built on a linear additive model. Formally, the hidden activation $\phi_{\beta}$ of a given sample $\beta$ are segmented into concept contributions/unique features $\phi_{\beta}^l$. Thus, the value function corresponding to the underlying Shapley values is given by $v_{\beta}(S) = \sum_{l \in S} \phi_{\beta}^l \cdot w$ (linear in $\phi^l$) for $S \subseteq \{1, \ldots, n_c + 1\}$.

spatial pooling. As for the concept activation maps, we use spatial upsampling to map $r_{xy}^l$ back to the input space and obtain *concept relevance heatmaps*. Since upsampling preserves the completeness relation, these decompose the *local relevance maps*, commonly referred to as class activation maps (CAMs) (Zhou et al., 2016), $r_{x,y} = \frac{1}{WH}\phi_{xy}^\beta \cdot \boldsymbol{w}$ into concept contributions.

**Global relevance and completeness score** Next, we establish a *global* (model-wide) concept relevance score, which measures the extend to which the concepts explain/cover the overall prediction strategy of the model. Recall, that all $\boldsymbol{c}_j^l$ defined above represent a basis for the feature space $\mathbb{R}^F$. Hence, we can directly decompose the weight vector $\boldsymbol{w}$ into (analogously to Equation (2))

$$\boldsymbol{w} = \sum_{l=1}^{n_c+1} \sum_{i=1}^{d^l} w_i^l \boldsymbol{c}_i^l \equiv \sum_{l=1}^{n_c+1} \boldsymbol{w}^l, \tag{4}$$

where $\boldsymbol{w}^l = \sum_{i=1}^{d^l} w_i^l \boldsymbol{c}_i^l$ and by construction, $\boldsymbol{w}^l \cdot \boldsymbol{w}^\perp = 0$ for $l = 1, \dots n_c$. In this case, we have

$$|\boldsymbol{w}|^2 = |\boldsymbol{w}^\perp|^2 + |\sum_{l=1}^{n_c} \boldsymbol{w}^l|^2 = \sum_{l=1}^{n_c+1} |\boldsymbol{w}^l|^2 + \sum_{l,k=1,l\neq k}^{n_c} |\boldsymbol{w}^l||\boldsymbol{w}^k|\cos(\angle(\boldsymbol{w}^l, \boldsymbol{w}^k)) \tag{5}$$

The first equality allows us to define

$$\eta(\{C^l\}) = 1 - |\boldsymbol{w}^\perp|^2/|\boldsymbol{w}|^2 \tag{6}$$

as a *completeness score* (fraction of $\boldsymbol{w}$ which is explained by all concepts $\{C^1, \dots, C^{n_c}\}$) with respect to a given class. To the best of our knowledge, we are the first to introduce a concept completeness score directly based on model parameters. Previous work (Yeh et al., 2020) defined a related measure based on model accuracy. Later, we will fix $\eta$ to define the number of concepts $n_c$. Note, that for an orthonormal basis (e.g., MCD-SSC-ortho, see below) the second term in Equation (5) (cosine) disappears. Then $|\boldsymbol{w}^l|/|\boldsymbol{w}|$ can be directly interpreted as (global) concept relevances, which sum up to the previous completeness score over all concepts. Further, the angles in Equation (5) are lower- and upper-bounded by the corresponding minimal or maximal principal angles[5] between the two corresponding subspaces, i.e., $\theta_{\min}^{kl} \equiv \min_m \theta_m^{kl} \leq \angle(\boldsymbol{w}^k, \boldsymbol{w}^l) \leq \max_m \theta_m^{kl} \equiv \theta_{\max}^{kl}$. This means we can lower- and upper-bound $|\boldsymbol{w}|^2$ by

$$\sum_l^{n_c+1} |\boldsymbol{w}^l|^2 + \sum_{l,k=1,l\neq k}^{n_c} |\boldsymbol{w}^l||\boldsymbol{w}^k|\cos(\theta_{\max}^{lk}) \leq |\boldsymbol{w}|^2 \leq \sum_l^{n_c+1} |\boldsymbol{w}^l|^2 + \sum_{l,k=1,l\neq k}^{n_c} |\boldsymbol{w}^l||\boldsymbol{w}^k|\cos(\theta_{\min}^{lk}). \tag{7}$$

Obviously, the lower and upper bound coincide in the case of orthogonal subspaces. This implies, that the $|\boldsymbol{w}^l|$ are also informative in the non-orthogonal case, provided the principal angles between the different subspaces are given. This highlights the intricate connection between (global) relevances and the geometry in feature space, i.e., the relative orientation of the concept spaces (specified via principal angles between pairs).

Finally, we briefly comment on the applicability of our approach for local and global concept relevances via Equation (3) and Equation (4). In the form described above it can be used for any model with a linear layer as final layer, potentially preceded by a global pooling layer, if one aims to spatially resolve the relevances instead of considering only pooled feature vectors. This latter category covers a broad range of modern CNN architectures such as ResNets, Inception-based models but also vision transformers, that do not base their prediction on a CLS token, such as Swin transformers (Liu et al., 2021). We envision, that our approach is even applicable, in approximate form, to other feature layers apart from the final hidden layer if one locally approximates the remainder of the model by a linear model, similarly as it is done by Ribeiro et al. (2016) or by Selvaraju et al. (2020) to generalize (Zhou et al., 2016).

## 2.4 Alternative MCD variants

In Section 2.2, we describe our algorithmic choices for the two steps of concept discovery, namely SSC for clustering of feature vectors and PCA for basis construction. As a consequence of the modularity of the

---

[5]A formal definition of principal angles is given in Appendix B.

MCD framework, we can easily define alternative variants of MCD, that also serve for an ablation study later. To differentiate the original MCD flavor described in Section 2.2, we name it *MCD-SSC*. Replacing SSC with other clustering methods gives rise to the following two MCD variants:

- *MCD-kmeans* We consider k-means clustering directly applied to the features. Like SSC, it leads to multi-dimensional and in general non-orthogonal subspaces. However, the clustering algorithm does not include any information about the linear subspaces as desired clustering target.

- *ICE/MCD-PCA* We consider PCA applied to the feature vectors directly. This corresponds to the concept discovery algorithm considered by ICE (Zhang et al., 2021). Note, that this approach already encompasses the basis identification step and directly leads to one-dimensional, orthogonal subspaces by construction.

MCD-SSC does not assume that two different subspaces $C^l$ and $C^m$ are orthogonal, as there is no mechanism that promotes this during model training. Still, concept orthogonality could be enforced through the use of dedicated orthogonal subspace clustering methods (Rahmani & Atia, 2017a), however, at the potential cost of slightly sub-optimal subspace clusters (Rahmani & Atia, 2017b). Alternatively, this could be implemented by sequentially rotating each identified subspace into the orthogonal complement of its predecessors. The latter leads to the last MCD flavor:

- *MCD-SSC-orth* We construct orthogonal subspaces from those discovered by MCD-SSC in an iterative fashion. Starting with an empty set, we explore the effect of adding one of the subspaces on the completeness defined in Equation (6) and choose the one that leads to the largest increase. Iteratively, we consider adding another subspace rotated into the orthogonal complement of the span of the subspaces in the set so far, again selecting the candidate that leads to the largest completeness increase.

Later, we find evidence that orthogonal are less faithful to the model than those that allow for arbitrary rotation.

## 3 Related Work

ACE (Ghorbani et al., 2019) uses a superpixel segmentation algorithm and k-means clustering to identify class-specific concept candidates for TCAV (Kim et al., 2018). The concept discovery scheme of ACE has several shortcomings: The segmentation into candidate concept patches is model-independent and thus, segments are not necessarily meaningful as perceived by the model. To enable clustering of intermediate CNN activations, segments are resized and mean padded to the original input shape. This leads to artificial, off-manifold samples with potentially distorted aspect ratios and discards the overall scale information. Finally, ACE relies on multiple heuristics to discard segments/clusters both before and after k-means clustering. In contrast, MCD is coherently based on hidden model representations without relying on additional pre- or post-processing. Similar limitations apply to methods that rely on ACE-discovered labeled concepts, like (Li et al., 2021), which uses Shapley values for concept importance, and (Wu et al., 2020), which occludes particular neurons for neuron-wise relevances and transforms them into concept importances via concept classification. Recently, Crabbé & van der Schaar (2022) proposed a generalization of TCAV by invoking the kernel trick, which generalizes the concept definition towards non-linear structures. However, unlike MCD, it does not allow quantifying the relevance of a concept towards the model prediction and can only verify predefined concepts instead of discovering them.

ICE (Zhang et al., 2021) defines concepts as directions in feature space. Technically, this is achieved via dimensionality reduction techniques applied to concatenated flattened feature maps. ICE measures the importance of its class-wise concepts using TCAV. Interestingly, ICE introduces the notion of a concept weight, which is analogous to our concept relevances on the logit layer. However, they do not consider spatially resolved concept relevance heatmaps and only address the special case of single-dimensional subspaces. Given these restrictions, ICE can be seen as a special realization of the MCD framework, which uses dimensionality

reduction methods like PCA as clustering algorithms. Other methods learn concept vectors and a mapping to feature space either for all classes simultaneously (ConceptSHAP (Yeh et al., 2020)) or for each class separately (MACE (Kumar et al., 2021), PACE (Kamakshi et al., 2021)). ConceptShap, MACE and PACE all use additional regularizers to enforce concept dissimilarity. In contrast, MCD restricts the concept discovery process as little as possible. Importantly, each method above defines a custom measure for concept importance, which is based on approximations of the original model. In contrast, the local and global concept relevance within MCD is solely based on the original model parameters. Other approaches (Chormai et al., 2022) use a concept definition similar to ours but use information from external attribution methods as well as orthogonality constraints to restrict the discovered concepts, whereas MCD works without such restrictions.

There is a complementary line of work of frameworks that try to identify concepts associated with particular neurons in hidden CNN representations, in conjunction with (Bau et al., 2017) or without (Achtibat et al., 2022) special concept-annotated datasets. Network Dissection (Bau et al., 2017) investigates the alignment of human-understandable concepts and particular single hidden features (neurons). Net2vec (Fong & Vedaldi, 2018) extended this by allowing concepts to be represented by combinations of neurons.

Lastly, there is a line of research that constructs inherently interpretable concept models by design with (Koh et al., 2020; Radenovic et al., 2022; Chen et al., 2020; Marconato et al., 2022; Zarlenga et al., 2022) or without relying on concept annotations (Chen et al., 2019). The objective of the former ante-hoc concept models is to discover concepts (Koh et al., 2020; Chen et al., 2020; Marconato et al., 2022; Zarlenga et al., 2022) in feature space in order to identify them with known factors from the underlying data-generating process. In contrast, the objective of MCD is to recover the true concepts learned by an arbitrarily trained model, which do not have to align with concepts underlying the data-generating process. Even though technically feasible (Chen & Feng, 2012), we do not equip MCD with a mechanism that enforces identifiability with known concepts, e.g., through weak concept supervision. There is a crucial difference in enforcing concept interpretability in the sense of identifiability with known concepts between ante-hoc and post-hoc approaches. Regularizing concept interpretability of post-hoc explanations might obfuscate the explanation and make the model appear more interpretable than it actually is. Concerning the latter category of concept models, our approach is best comparable with (Chen et al., 2019), as both can be reduced to a linear model operating on concepts that can be characterized via prototypes. We stress the essential difference, that our approach does not require retraining (with special training objectives) but is an interpretable reformulation of the original model.

## 4    Results

We carry out our experiments on ImageNet (Deng et al., 2009). As model architectures, we consider ResNet models (He et al., 2016) using original weights as provided by *torchvision* and updated weights as provided by *timm* (Wightman, 2019) with an improved training procedure (Wightman et al., 2021). We also present results for a swin vision transformer (Liu et al., 2021), again using weights provided by *timm* (SwinS3base224). In the following, we will refer to these models as ResNet50, ResNet50v2 and, Swin-T, respectively. We base all our experiments on images from a diverse selection of ten ImageNet classes, which roughly align with CIFAR10 classes[6]

### 4.1    Completeness arithmetic

First, we provide a concrete example for an MCD explanation and showcase its completeness relation introduced in Section 2.3 (*benefit 3*). To this end, Figure 3 shows an MCD-SSC explanation of a ResNet50v2 prediction for a sample of the police van class in ImageNet. The number of concepts was chosen such that the completeness measure in Equation (6) reaches $\eta = 0.5$. The three information components of the explanation all provide complementary information:

---

[6]namely (airliner, beach wagon, hummingbird, siamese cat, ox, golden retriever, tailed frog, zebra, container ship, police van)

(1) *Concept relevance heatmaps* show the alignment of a feature vector component $\phi^l$ associated with concept $C^l$ and the weight vector of a specific class. Roughly speaking, this alignment indicates how typical the network perceives the particular instantiation of the concept for the class under consideration. Applying mean pooling leads to a corresponding decomposition of the class logit under consideration (up to the bias term) into contributions corresponding to different concepts. This demonstrates the completeness relation on the level of concept relevance heatmaps as well on the level of logits, which represents a unique feature of the MCD framework. Interestingly, for the explanation in Figure 3, only the orthogonal complement concept contributes negatively to the class logit. The contributions of the first two concepts clearly dominate the class logit.

(2) *Concept activation maps* have positive scores showing how much a particular feature vector aligns with a specific concept subspace. These maps identify input regions where the concept is highly expressed. We color-code concept activation maps as a transparent overlay over the image where transparent regions indicate high activation. To guide the eye, we also include a yellow contour line at a threshold value of 0.5 and a white one at a value of 0.4.

(3) *Concept prototypes* allow characterizing a concept subspace through examples. Here, we display the concept activation maps of three test set samples that show the highest activation with the given concept. In many cases, an intuitive meaning of a concept can be inferred most easily from these samples and numerous previous approaches present concepts in this way (Zhang et al., 2021; Achtibat et al., 2022; Yeh et al., 2020). In case of the explanation in Figure 3, this could be windows/livery, livery, blue lights, building, tires (and the orthogonal complement covering mainly the background). In addition, we also indicate the global concept relevances for the different concepts according to Equation (4).

At this point, we deem it worthwhile discussing the complementary nature of concept relevance maps and concept activation maps:

- Concept activation maps facilitate concept identification as they do not entangle concept activation and model prediction. More precisely, concept activation maps provide insight into the structures present in the feature space, including those that may not directly contribute to the prediction. This is particularly evident in the case of the orthogonal complement, which is slightly activated in Figure 3 but has no positive relevance.

- Concept the positive parts of the concept relevance maps correlate with the activation maps: Among all test set samples, we find a mean Pearson correlation of 0.45 for the concepts of the CIFAR10 classes between the positive part of each concept relevance map and the corresponding concept activation map (for MCD-SSC and ResNet50v2). This result serves as a sanity check and confirms that concept relevance is high in sample areas where the respective concept is strongly activated. Importantly, the negative parts in the concept relevance maps represent additional information.

In summary, the sample in Figure 3, is classified as a police van mainly due to its windows/livery, which are perceived as typical for the class by the network and are also the most relevant concept for the class globally. Further, all other concepts are expressed in the sample and contribute positively, except for the orthogonal complement. Thus, we can confirm that MCD-SSC concepts indeed capture and focus on the relevant model reasoning structure.

## 4.2 Empirical evaluation

We compare MCD with sparse subspace clustering (MCD-SSC), MCD with alternative clustering, and previous methods listed in Table 1 in terms of (1) faithfulness (*benefit 1*) and (2) conciseness (*benefit 2*) of the explanations.

### 4.2.1 Comparing faithfulness via concept flipping

In order to compare the methods in Table 1 in terms of faithfulness, we invoke the Smallest Destroying Concepts (SDC) benchmark as proposed in (Ghorbani et al., 2019) and (Wu et al., 2020). For concepts that

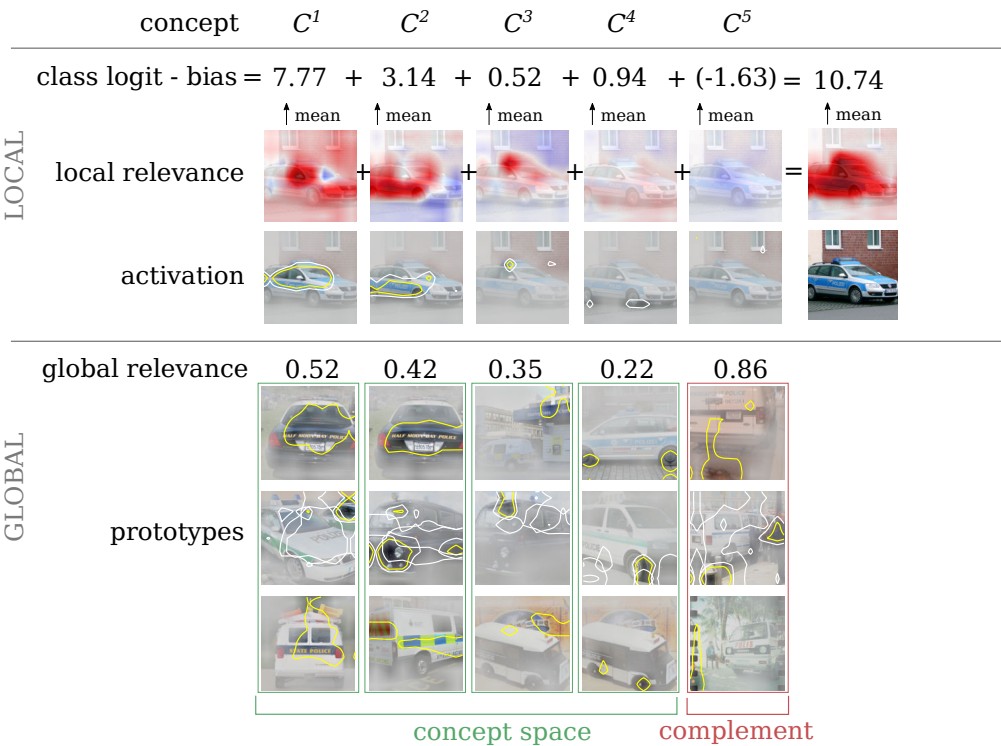

Figure 3: Completeness relation for the police van class in ImageNet. Concepts are discovered via MCD-SSC for ResNet50v2. The number of concepts is chosen such that the completeness score reaches $\eta = 0.5$. We distinguish between local (sample-specific) and global properties (characterizing a set of samples). Locally, we consider *concept relevance maps*, which quantify the spatially resolved contribution of a concept to the prediction. These satisfy a *completeness relation*, as explicitly shown in the first line. *Concept activation maps* provide complementary information and indicate how much a concept is activated depending on the spatial location in the sample. Globally, the overall relevance of a particular concept is quantified by the *global relevance* scores. Finally, we also present concept prototypes (concept activation maps of the most strongly activated samples) to characterize a particular concept.

Table 1: Summary of concept discovery methods considered in this work.

| Method | Multi-dim. | Arbitrary orientation |
|---|---|---|
| MCD-SSC | ✓ | ✓ |
| MCD-SSC-ortho | ✓ | ✗ |
| MCD-kmeans | ✓ | ✓ |
| ICE/MCD-PCA (Zhang et al., 2021) | ✗ | ✗ |
| ACE (Ghorbani et al., 2019) | ✗ | ✓ |

reflect the model's actual reasoning structure in feature space and *faithful* concept relevance scores, SDC should show a sharp decline of the model accuracy with the number of flipped concepts.

To evaluate SDC, we subsequently remove concepts, as represented by concept masks in input space, in order of their sample-wise (local) relevance starting from high to low. To inpaint the removed segments, we use a classical imputation algorithm (Bertalmio et al., 2001), which leads to comparably realistic imputed images. Thus, the model is evaluated on-manifold in contrast to imputing with gray patches as often done in the literature (Samek et al., 2017). For similar reasons, we avoid the Smallest Sufficient Concepts (SSC)

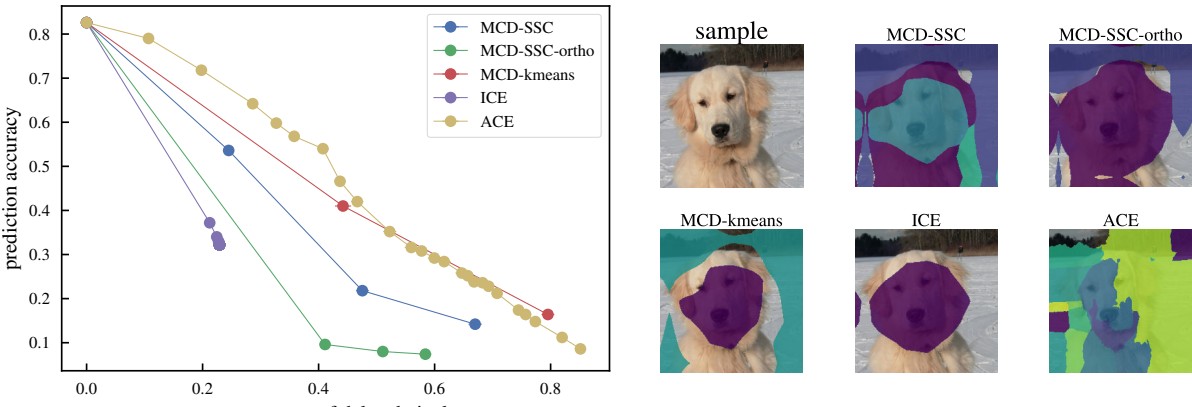

Figure 4: Left: Concepts are flipped one at a time in descending order of local concept importance/TCAV score, respectively. We measure the decline in model accuracy and show the mean accuracy across CIFAR10 classes against the fraction of deleted pixels. Meaningful concept discovery and quantification methods are supposed to show a sharp decline in this figure, but the decline should not happen after flipping only a single concept (i.e. the whole object). Right: Qualitative comparison between hard concept assignments.

benchmark, which would require high-quality imputation algorithms to avoid evaluating the model far from the data manifold. We obtain concept masks, i.e., hard concept assignments, in input space by taking the argmax of the corresponding concept activation maps over all concepts including the orthogonal complement. After the argmax operation, we disregard (do not remove) the orthogonal complement during the SDC experiments. For each concept mask we obtain local relevance scores by pooling the corresponding concept relevance heatmaps over the respective regions. This provides concept masks in input space which are ordered according to their importance. ACE does not provide a measure of per-sample concept relevance. Therefore, we revert to the order of their (global) TCAV scores after discarding concepts where statistical testing in comparison to random input samples fails to stay below $p = 0.05$. In contrast to previous studies (Ghorbani et al., 2019; Wu et al., 2020), we report the model performance depending on the fraction of occluded pixels, which is essential for comparability since the segment size varies between different approaches. In order to show a meaningful average of the samples across all classes we flip only as many concepts as are present for the class with minimum number of concepts $n_c$ for each method. We base our evaluation on the CIFAR10 classes described above and work with the ResNet50v2 model, for which we extract concepts from the last hidden layer. For all methods within the MCD framework, we fix the number of concepts in a class-dependent way such that we reach a completeness score of $\eta = 0.5$.

In the left panel of Figure 4, we show the results of the SDC experiment. As mentioned above, a meaningful concept discovery and quantification method should show a sharp decline in Figure 4. However, in principle, the sharpest decline is achievable via assigning the complete object to a single concept. Such a concept would simply highlight the entire relevant region, i.e., this would not provide any insights beyond those that can be inferred by standard (non-concept-based) attribution methods such as LRP (Bach et al., 2015), PredDiff (Blücher et al., 2022) or Shapley values (Lundberg & Lee, 2017). The SCD results show, that ICE/MCD-PCA and MCD-SSC-ortho consistently detect only a single relevant concept, as the accuracy curve stagnates after flipping the first concept. Thus, ICE/MCD-PCA and MCD-SCC-ortho, counteract the benefits of concept-based explanations. Since both approaches rely on orthogonal concepts, this constraint seems unfitted for a fine-grained analysis of related but distinct characteristics in feature space. Among the remaining algorithms, MCD-SSC shows the strongest decline as compared to MCD-kmeans and ACE, hence its discovered concepts are the most faithful.

To provide a qualitative impression of the concept relevance heatmaps across methods, we show them together with concept activation maps for a selected sample of the golden retriever class (one of the CIFAR10 classes)

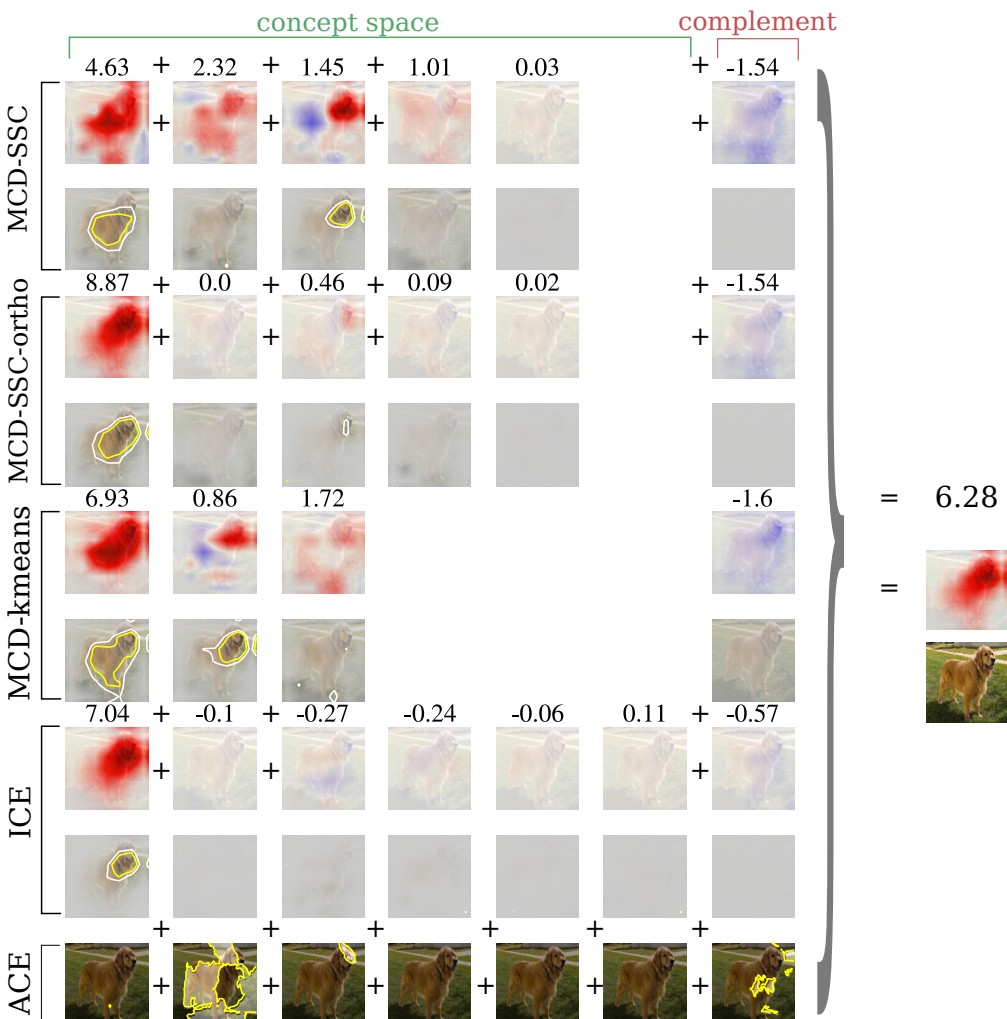

Figure 5: Concept heatmaps and activation maps for ResNet50v2 and a randomly chosen sample from the golden retriever class in ImageNet. The number of concepts is chosen such that the completeness score reaches $\eta = 0.5$. Concepts are ordered from left to right according to global concept relevance. Concept heatmaps are titled by the pooled local concept relevance that sums to the prediction logit minus the bias. For ICE, we only show the first six out of 142 and for ACE the first six out of 25 concepts. For ACE, no complement exists. For MCD-kmeans, relevance is distributed over all three concepts and also four of the five MCD-SSC concepts have notable relevance. In contrast, among the MCD flavors with orthogonal concepts (MCD-SSC-ortho and ICE), only one concept notably contributes to the prediction.

in Figure 5, and more equivalent results for other classes in Figures 7 and 8.[7] In Figure 4 (right panel), we also show hard concept assignments for an example image of the golden retriever class, which form the basis of the concept flipping experiment described above. These visually support the findings of the concept flipping experiment. Most approaches only discover a single concept for the dog (apart from a potential genuine background concept). In particular, consider the concept assignments in Figure 4 on the right: Here, both orthogonal approaches do not distinguish between the dog head and fur parts on the image. In contrast, the unconstrained MCD-SSC approach can successfully distinguish these two correlated regions in features space (incorporate two similar concepts) and shows the most fine-grained decomposition.

---

[7]Correspondong prototypes can be found in Figures 9 to 11.

To summarize the results of the concept flipping experiment, our general MCD definition leads to the most faithful concepts, as the two unconstrained MCD flavors (MCD-SSC and MCD-kmeans), show the steepest descent among all methods without reverting to the non-informative solution of a single relevant concept.

### 4.2.2 Conciseness of explanations

For an accessible explanation, it is desirable, to explain the model reasoning with as few meaningful concepts as completely as possible, i.e. to deliver concise concept explanations. To compare the conciseness of concept explanations we measure the number of concepts required to reach a certain completeness score $\eta$, i.e. how many concepts are necessary to cover the whole relevant feature space. Again, there is a trivial solution, namely leveraging more and more dimensions to cover the majority of feature space via a single concept. Therefore, we additionally evaluate the average subspace dimension $d_l$ and the mean (scaled) Grassmann distance $\Delta_c^{kl}$, as defined in Equation (10), between all concept pairs $(k, l)$ within one class $c$ to quantify how dissimilar two concepts are. [8] In summary, we argue that concepts should be concise (small $n_c$), but dissect the feature space into meaningful building blocks of model reasoning. While the latter is difficult to quantify, we argue that there is a trade-off between (1) covering feature space with very few concepts of high dimensionality and potentially small distance vs. (2) dissecting it into a high number of concepts with small dimensionality (extreme case: one-dimensional). To support this reasoning, we also inspect the visual impression of concepts for a selection of classes. Again, we base our evaluation on the CIFAR10 classes and concepts for the last hidden layer of the ResNet50v2 model. For all methods within the MCD framework, we fix the number of concepts in a class-dependent way such that we reach a completeness score of $\eta = 0.5$.

We list the number of concepts $n_c$ that is required to reach a completeness score of $\eta = 0.5$ and $d_l$ in Section 4.2.1. To provide a visual comparison of the concepts discovered by these methods, we show concept activation maps of prototypes for basketball, golden retriever and airliner class in ImageNet in Figures 9 to 11 and judge how broad they appear in input space. MCD-kmeans discovers the smallest number of concepts with the highest mean concept dimensionality of 74.7 and the smallest inter-concept distance $(\text{mean}(\Delta_c^{kl}) = 0.83)$ among all methods. This is reflected in the visual appearance of the concept prototypes, which are visually broad and difficult to distinguish. MCD-SSC discovers on average 4.8 concepts with a 41% smaller mean concept dimensionality of 44.2. Visually, concepts are medium broad and are easier to distinguish in input space than for MCD-kmeans, which is reflected in a higher inter-concept distance of 1.19. When requiring orthogonality the Grassman angle is fixed to $\text{mean}(\Delta_c^{kl}) = \pi/2 = 1.57$ (MCD-SSC-ortho and ICE). For orthogonal concept one concept is medium broad in input space while all others are almost not activated. Most likely, the orthogonality constraint hinders the concepts to reflect a natural similarity between certain concepts. This aligns with the conclusions drawn from the SDC benchmark. Most notably, to achieve a comparable model faithfulness (completeness score of $\eta = 0.5$) 30 times more one-dimensional ICE concepts than multi-dimensional MCD concepts are required, meaning this method delivers concept explanations that are not concise. Intuitively, a single concept is split up into several concepts, which is also reflected in their weak activation on test set samples. Lastly, the visual impression of ACE concepts is fixed by the choice of the superpixel algorithm. While ACE concepts are all one-dimensional, they do not provide a mechanism to quantify how complete they are, thus we cannot quantify $n_c$ required to reach a completeness of 50%. As an overall summary, MCD-SSC is superior in dissecting the feature space into enclosed and meaningful concepts.

### 4.3 Use case: MCD concepts reveal differences in classification strategies between model architectures and training procedures

Finally, we showcase how MCD can unravel different classification strategies depending on the model architecture (ResNet50 vs. Swin-T) and the training strategies (ResNet50 vs. ResNet50v2), most notably the fact that the ResNet50v2 was trained using a multilabel loss. The test accuracies for the subset of CIFAR10-classes are 0.80 (ResNet50), 0.84 (ResNet50v2) and 0.86 (Swin-T). Here, we focus on MCD-SCC and, as before, restrict ourselves to concepts in the last hidden feature layer. First, we compare the discov-

---

[8]We use a scaled version of the original Grassmann distance that aggregates the principle angles (in radian) between two subspaces, for which $0 \leq \Delta_c^{kl} \leq \pi$. Two special cases are $\Delta_c^{kl} = 0$, meaning that subspace bases vectors are perfectly aligned, and $\Delta_c^{kl} = \pi/2$, meaning that they are orthogonal.

Table 2: Summary of concept discovery methods considered in this work in comparison to prior work from Zhang et al. (2021) (ICE/MCD-PCA) and Ghorbani et al. (2019) (ACE). We measure average subspace dimension $d^l$ and the number of concepts $n_c$ that is required to reach a completeness score of $\eta = 0.5$ for ResNet50v2 on the CIFAR10 classes. A small number of relevant concepts $n_c$ is desirable since this summarizes the complete model into an accessible and meaningful format. Here, multi-dimensional concepts have an advantage. Additionally, we evaluate the mean (scaled) Grassmann distance $\Delta_c^{kl}$, see Equation (10), between all concept pairs $(k, l)$ within one class $c$ to quantify the distinctness between concepts. The visual inspection is based on prototypes of the basketball, golden retriever and airliner class concepts in Figures 9 to 11. Medium broad and distinct concepts are the most informative.

| Method | mean($d^l$) | mean($n_c$) | mean($\Delta_c^{kl}$) | Visual inspection |
|---|---|---|---|---|
| MCD-SSC | 44.2 | 4.8 | 1.19 | medium broad |
| MCD-SSC-ortho | 44.2 | 4.8 | 1.57 | only one broad (rest narrow) |
| MCD-kmeans | 74.7 | 2.7 | 0.83 | very broad |
| ICE/MCD-PCA | 1 | 146.7 | 1.57 | only one broad (rest narrow) |
| ACE | 1 | n.a. | n.a. | medium broad |

ered concepts between the models by the activation maps of concept prototypes for the beach wagon class of ImageNet in Figure 6. We fix the number of concepts to $n_c = 5$. For Swin-T, we only apply a spatial upsampling of the concept activation maps from the feature to the input space to $14 \times 14$ in order to account for the $16 \times 16$ patch tokenization. We find that ResNet50 concepts, which could roughly be identified as (car body, windows, car roof, wheels, street), are more narrow than the expression of Swin-T and ResNet50v2 concepts. The latter are related to broader views of the car, such as concepts (1, 2, 4) for ResNet50v2 and concepts (1, 3) for Swin-T. Interestingly, ResNet50v2 concepts reach a much lower completeness score of $\eta = 0.49$ than ResNet50 ($\eta = 0.89$) and Swin-T ($\eta = 0.84$) for fixed $n_c = 5$. In Figure 6 we show the relation between the total concept space dimensionality, the number of concepts $n_c$ and the completeness score $\eta$ across the CIFAR10-classes. Even for $n_c = 30$, the ResNet50v2 concepts have a lower $\eta$ than those of the ResNet50 for $n_c = 3$, although the former covers already a much larger part of the concept space. These observations support the statement that feature space of the ResNet50v2 exhibits a comparably richer structure than ResNet50 . Thus, MCD-SSC concepts can reveal interesting differences in the character of the feature space as a consequence of two different training procedures for the same architecture. Interestingly, the dependence of $\eta$ on $n_c$ for the concepts between two models with different architectures, ResNet50 and Swin-T, is quite similar. This also aligns with the visual appearance of the concepts.

To summarize, Swin-T and ResNet50 build on broader and more versatile concepts. In comparison, ResNet50v2 builds on more narrow and thus specific concepts for its classification strategy. These broad concepts are not unexpected for a transformer architecture like Swin-T with coarse self-attention windows, but a rather surprising finding for ResNet50 in comparison to ResNet50v2.

## 5 Summary and Discussion

In this work, we put forward MCD, a general framework for concept discovery based on the hidden representation of a trained deep neural network. Unlike prior work in the field, we propose a general concept definition (incorporating previous approaches) as multi-dimensional linear subspaces without restricting to single directions or enforcing orthogonality between subspaces. We use concept activation maps to visualize concepts in input space. Considering the final hidden layer representation, we can reformulate the original model as a linear classifier acting on linear concept subspaces without the need to retrain with a special objective. This leads to a completeness relation, i.e., a natural decomposition of class logits into contributions corresponding to specific concepts and allows to resolve their spatial importance in terms of concept relevance heatmaps. As a particularly suited realization of our framework, we put forward MCD-SCC, which

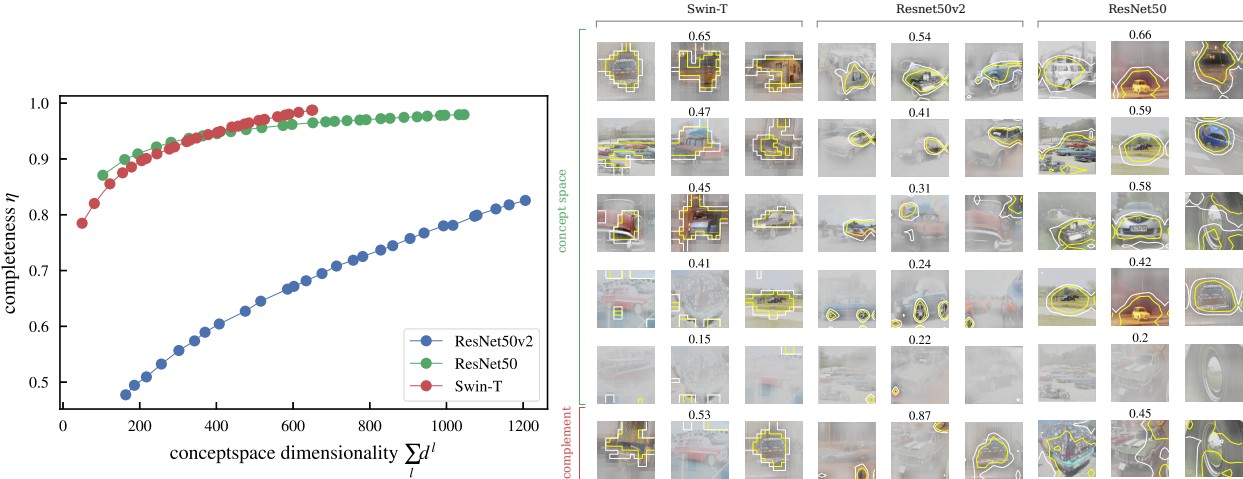

Figure 6: Left: Mean concept space completeness score $\nu$ for the CIFAR10 classes across architectures against the dimensionality of the union of all concept subspaces $\sum_l d^l$. The number of concepts can be inferred from the points on the line where the first point on each line corresponds to $n_c = 3$ and the last one to $n_c = 30$. ResNet50v2 shows a much lower completeness score at roughly the same $n_c$ and $\sum_l d^l$ as ResNet50. The feature space dimensionality is $F = 2048$ for ResNet50(v2) and $F = 768$ for Swin-T. Right: We show MCD-SSC concept activation maps for concept prototypes for ResNet50, ResNet50v2 and Swin-T and the beach wagon class in ImageNet. We fixed the number of concepts to $n_c = 5$. In this way, ResNet50v2 reaches $\eta = 0.49$, ResNet50 $\eta = 0.89$ and Swin-T $\eta = 0.84$. Each row shows a single concept and is titled by its global concept importance. The last row shows the orthogonal complement of the concept space.

relies on sparse subspace clustering for concept discovery. Based on qualitative and quantitative insights, we show the superiority of MCD-SCC over other MCD flavors that build on traditional clustering algorithms.

We showcase the ability of MCD via discriminating between hidden representations obtained from different model architectures and training strategies. This paves the way towards further novel use-cases for MCD concepts such as gaining insights in the natural sciences, e.g., identifying sub-classes of cancerous cells in histopathology or summarizing model behavior beyond single examples and thereby systematically discovering model biases. MCD prioritizes faithfulness of concepts over identifiability with known human concepts by not including any interpretability-enforcing regularizers that could obfuscate the original structures learned by the model. In this way, we can guarantee to cover the original model reasoning structure, which is crucial for auditing models or scientific discovery use cases. However, this might obfuscate non-expert human users at first sight, since deep learning models will most likely not rely entirely on human-like features.

Code to reproduce our experiments is publicly available at `https://github.com/jvielhaben/MCD-XAI`.

## Acknowledgments

This work was supported by the German Ministry for Education and Research (BMBF) through BIFOLD (refs. 01IS18025A and 01IS18037A).

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

# A    SSC Algorithmic Details

**Concept-determining self-representation** We compute sparse self-representations $R$ for a random sub-collection of $n \leq N \cdot H \cdot W$ feature vectors $\{\boldsymbol{\phi}_{xy}^\alpha\}$ sampled from $S$. Here, the term self-representation refers to a coefficient matrix that expresses each sample as a linear combination of all other samples. More specifically, using the notation from (Elhamifar & Vidal, 2013), given the feature vectors $\Phi = [\boldsymbol{\phi}_1, \ldots, \boldsymbol{\phi}_n] \in \mathbb{R}^{F \times n}$, we identify a sparse coefficient matrix $\boldsymbol{R} = [\boldsymbol{r}_1, \ldots, \boldsymbol{r}_n] \in \mathbb{R}^{n \times n}$ such that

$$\boldsymbol{\phi}_j = \Phi \boldsymbol{r}_j \text{ where } r_{ii} = 0. \tag{8}$$

The particular kind of sparsity constraints that are imposed on Equation (8) and how it is optimized depends on the chosen SSC algorithm. Here, we use elastic net subspace clustering (You et al., 2016a), which is robust against noise and scales well for large sample sizes. In all our experiments, we fix the hyperparameter $\gamma$, which balances sparsity vs. robustness, to $\gamma = 10$. We confirmed that the results are not sensitive to variation of this parameter over a range of values from 5 to 50. As computation time for SSC is dependent on this parameter, we chose $\gamma$ such that this is minimized.

We remove outliers based on the $\ell_1$-norm as in (Soltanolkotabi & Candes, 2012), where we empirically fix the percentile threshold to 0.75 and re-fit the sparse self-representation for the remaining elements.

Another scalable alternative to the elastic net clustering is orthogonal matching pursuit (OMP)(You et al., 2016b), which is, however, not robust against noise and does not allow for outlier removal via thresholding. Finally, the original sparse subspace clustering method from (Elhamifar & Vidal, 2013) is robust against noise and outliers but does not scale to large datasets. The particular robustness and scalability properties make elastic net subspace clustering (with thresholding) an ideal choice for the first step of our concept discovery method.

**Spectral clustering** In a second step, we perform spectral clustering with the affinity matrix $W = |R| + |R^T|$, which encodes the similarity of two feature vectors according to their self-representations. We determine the number of clusters $n_c$ either via the largest gap in the spectrum of the Laplacian (Von Luxburg, 2007) or use a predetermined value. This step assigns every input feature $\boldsymbol{\phi}_i$ to a particular cluster $\mathcal{C}_1, \ldots, \mathcal{C}_{n_c}$ or to the set of outliers.

# B    Characterizing relations between subspaces by principal angles

In this section, we briefly review the definition of principal angles, which can be used to characterize the relation between two linear subspaces. The principal angles $\theta_i^{AB}$ (Jordan, 1875) $(i = 1, \ldots, \min(\dim A, \dim B))$ between two linear subspaces $A, B$, are defined recursively via

$$\cos \theta_i^{AB} = \max_{\boldsymbol{a} \in A, \boldsymbol{b} \in B} \frac{\boldsymbol{a}^T \boldsymbol{b}}{|\boldsymbol{a}||\boldsymbol{b}|} =: \frac{\boldsymbol{a}_i^T \boldsymbol{b}_i}{|\boldsymbol{a}||\boldsymbol{b}_i|}, \tag{9}$$

where the maximum is taken subject to the orthogonality constraints $\boldsymbol{a}^T \boldsymbol{a}_j = 0$ and $\boldsymbol{b}^T \boldsymbol{b}_j = 0$ for $j = 1, \ldots, i - 1$.

To quantify the similarity between two subspaces $\mathcal{A}$ and $\mathcal{B}$, we use a scaled version of their Grassmann distance Hamm (2008), which is defined as,

$$\Delta^{AB} = 1/\sqrt{\min(\dim A, \dim B)} \sqrt{(\theta_1^{AB})^2 + \ldots + (\theta_{\min(\dim A, \dim B)}^{AB})^2}. \tag{10}$$

This allows comparing the similarity of concepts within a given class or across classes regardless of the concept subspaces' dimensionality.

## C  Qualitative results

For a qualitative comparison between of the concept activation maps and relevance heatmaps between the methods in Section 4.2, we provide results for selected samples in Figures 5, 7 and 8. In Figures 9 to 11 we show the respective concept prototypes for all concept discovery approaches.

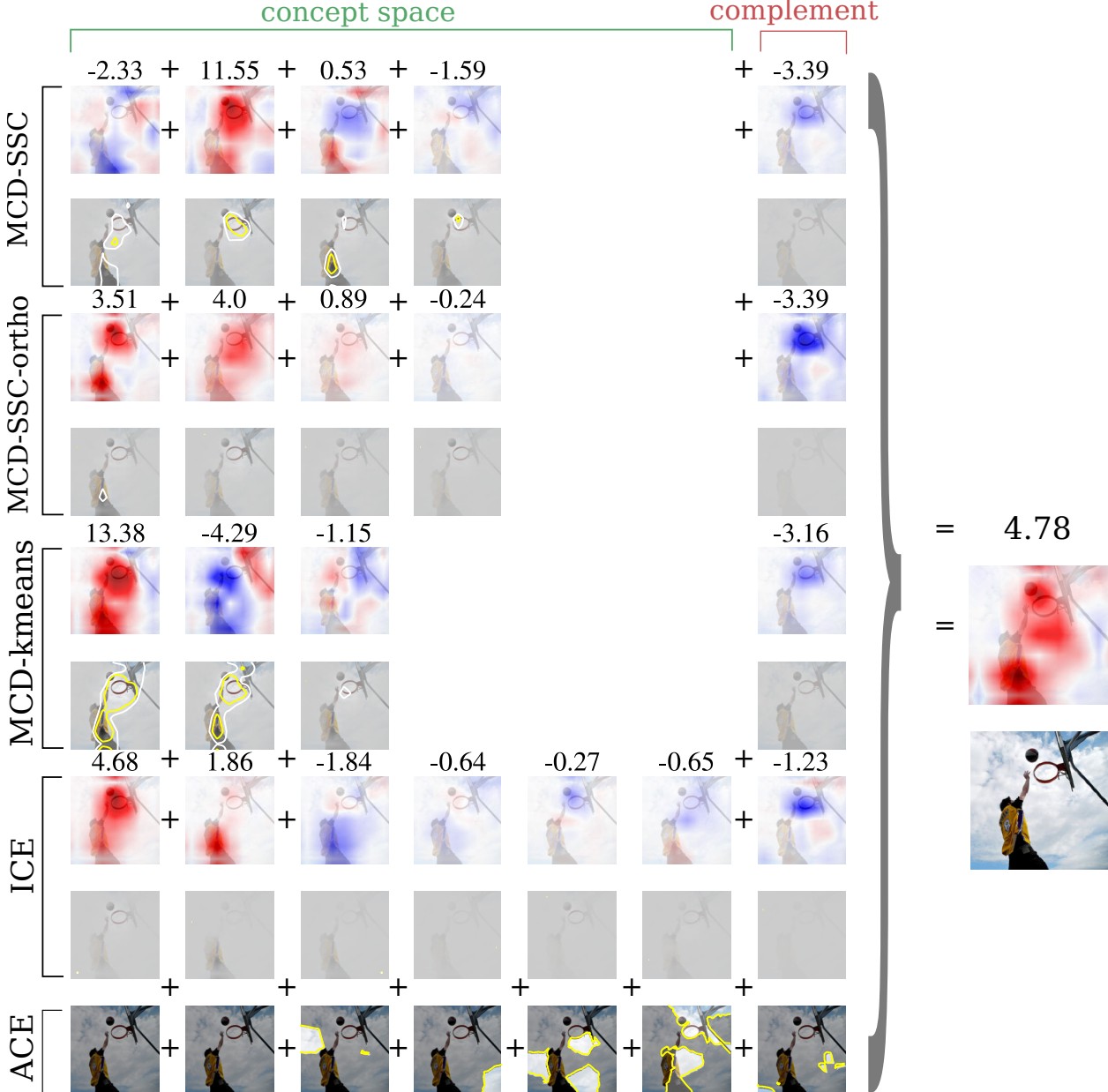

Figure 7: Concept heatmaps and activation maps for ResNet50v2 and a randomly chosen sample from the basketball class in ImageNet. The number of concepts is chosen such that the completeness score reaches $\eta = 0.5$. Concepts are ordered from left to right according to global concept relevance. Concept heatmaps are titled by the pooled local concept relevance that sums to the prediction logit minus the bias. For ICE, we only show the first six out of 105 and for ACE the first six out of 25 concepts.

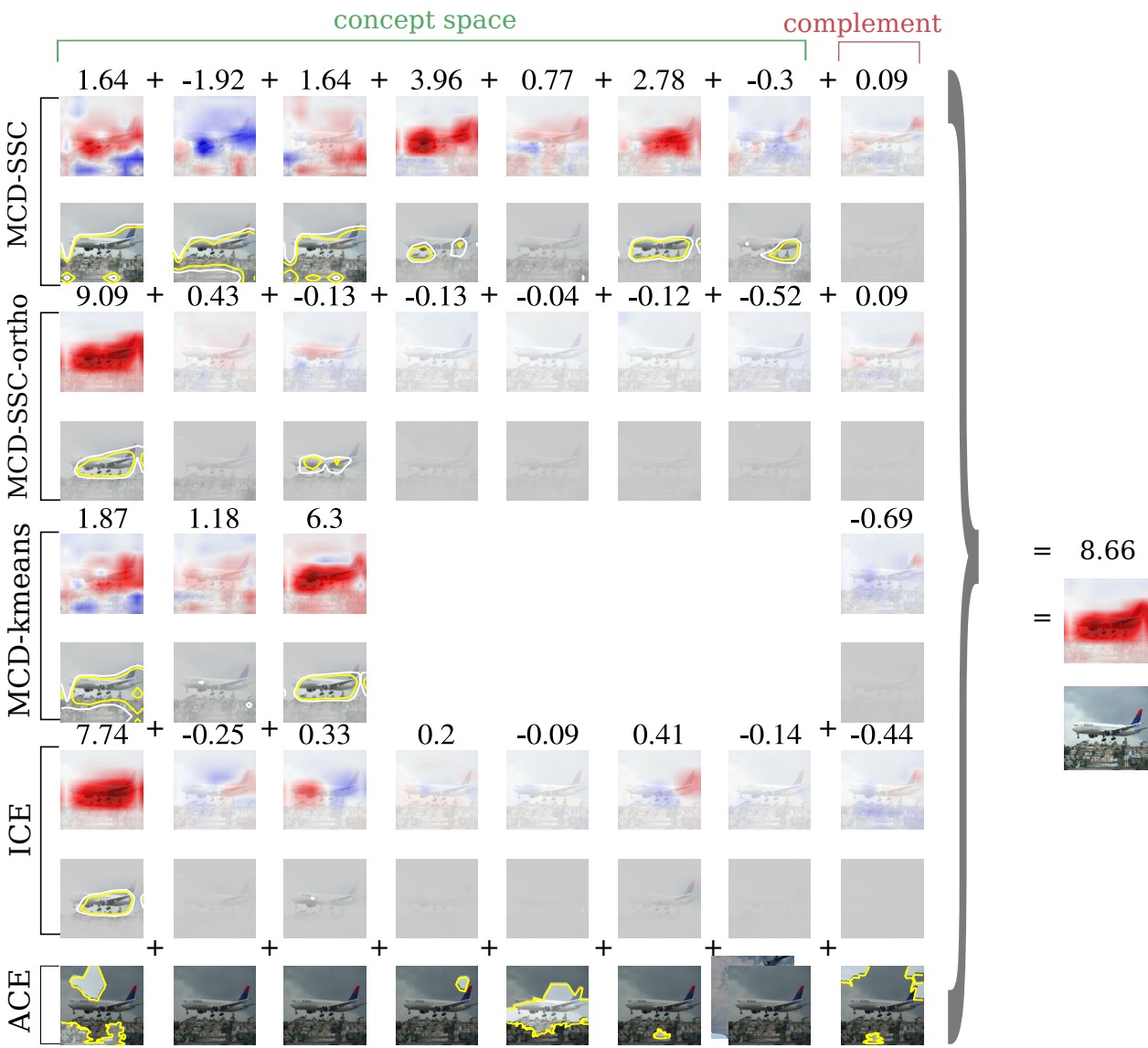

Figure 8: Concept heatmaps and activation maps for ResNet50v2 and a randomly chosen sample from the airliner class in ImageNet. The number of concepts is chosen such that the completeness score reaches $\eta = 0.5$. Concepts are ordered from left to right according to global concept relevance. Concept heatmaps are titled by the pooled local concept relevance that sums to the prediction logit minus the bias. For ICE, we only show the first seven out of 141 and for ACE the first seven out of 25 concepts.

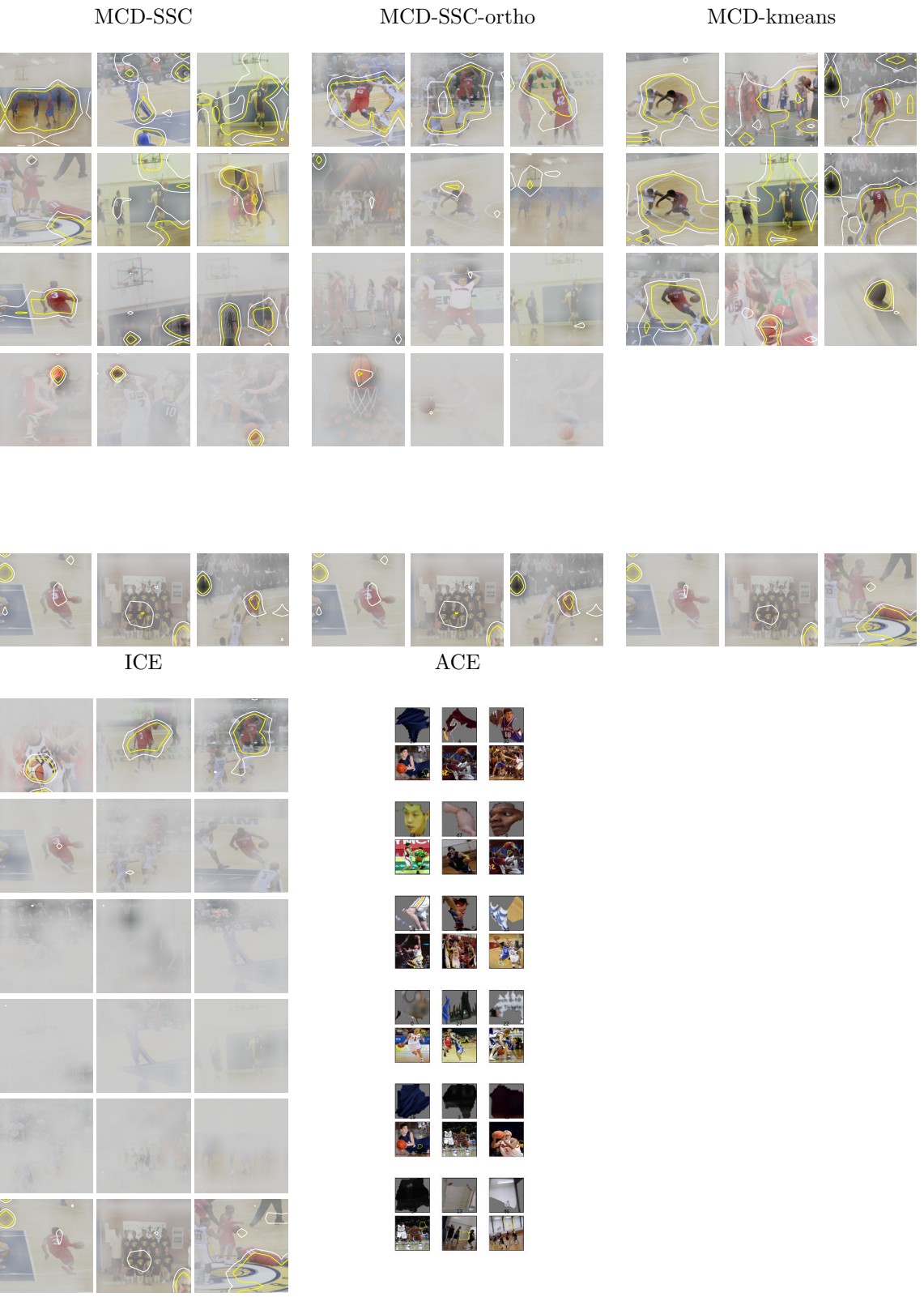

Figure 9: Concept activation maps for concept prototypes for basketball class of ImageNet. The last row shows prototype for the complement, except for ACE, where no complement exists. For ICE, we only show the first six out of 105 and for ACE the first six out of 25 concepts.

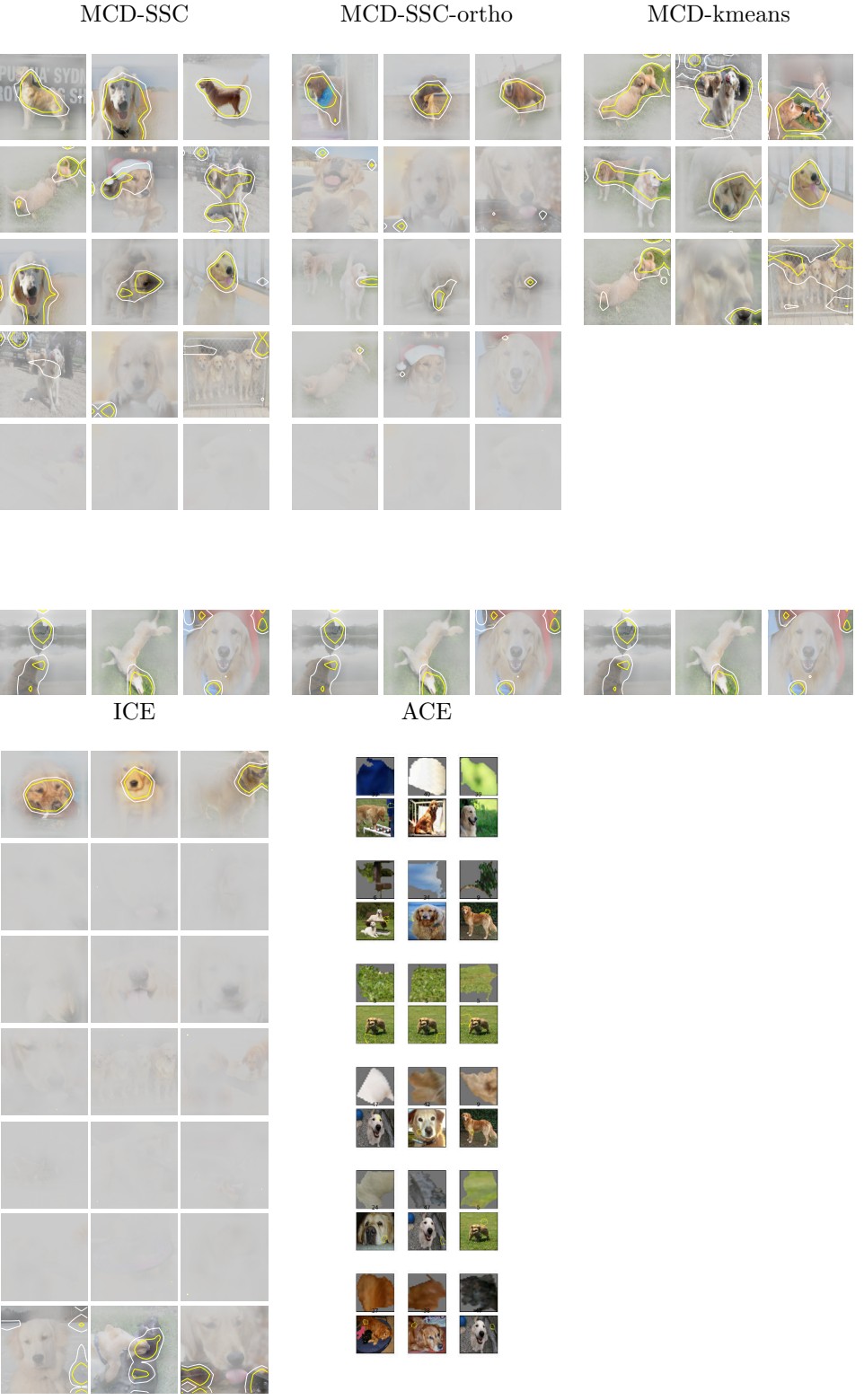

Figure 10: Concept activation maps for concept prototypes for golden retriever class of ImageNet. The last row shows prototype for the complement, except for ACE, where no complement exists. For ICE, we only show the first six out of 142 and for ACE the first six out of 25 concepts.

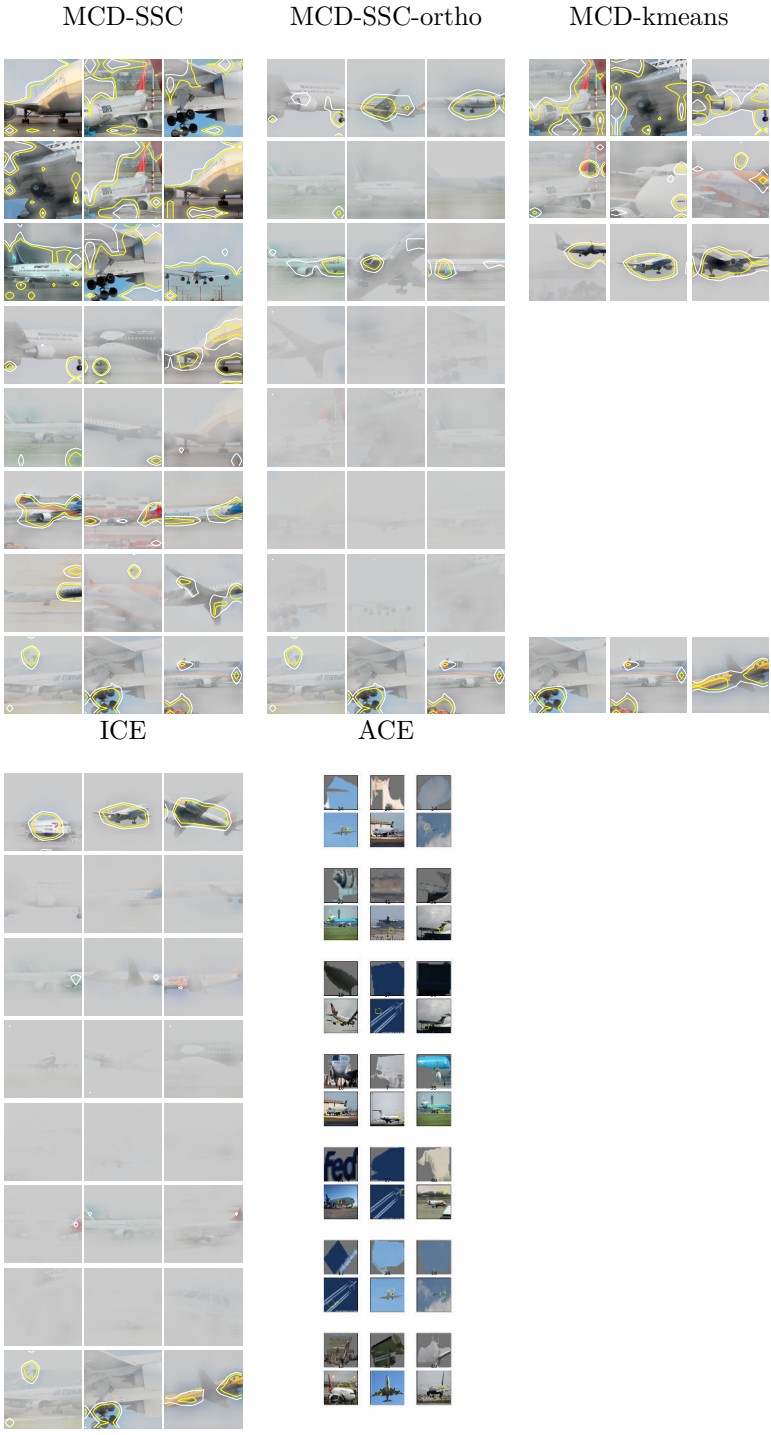

Figure 11: Concept activation maps for concept prototypes for airliner class of ImageNet. The last row shows prototype for the complement, except for ACE, where no complement exists. For ICE, we only show the first seven out of 141 and for ACE the first seven out of 25 concepts.

