# OpenReview forum: "Multi-dimensional concept discovery (MCD): A unifying framework with completeness guarantees"
_TMLR — Accepted by TMLR_

### Review · Reviewer_FPuj · 2023-02-13

**Summary Of Contributions:**

The authors introduce a novel post-hoc, concept based explainer for neural networks.

**Audience:**

Yes

**Broader Impact Concerns:**

As I mentioned, explanations extracted by *unsupervised* concept extraction methods always risk being potentially misinterpreted by stakeholders.  The authors should discuss this in the conclusion.  See:

 Chen et al., "Does the explanation satisfy your needs?: A unified view of properties of explanations"

for a reference.  This is a very recent overview of properties of explanations and mentions pitfalls in interpreting them.

**Claims And Evidence:**

Yes

**Requested Changes:**

*Writing*  English is good, and I could not spot any major linguistic mistakes.  Structure and clarity, however, should be improved.

- The introduction is far too detailed, the description of the method is not detailed enough.  I think the structure should be changed to improve clarity and flow.  Specifically,

 1) The technical paragraphs at the end of the introduction are very hard to follow at this stage (as the reader is not yet familiar with the intent of the various steps and with the techniques involves), and should be moved later, preferably between Sec 2.1 and 2.2.

 2) The schematic overview of the algorithm should be moved between to the beginning of Sec 2, and it should briefly outline the clustering, basis construction, and relevance computation steps, clearly stating what are the expected inputs and outcomes of the various steps *formally*.

 3) All algorithmic details should be kept in Sections 2.2+.  Intuition about important steps of the procedure are currently missing.  The authors should briefly describe:
 - What is subspace clustering?
 - How does SSC work?
 - Why is PCA enough to recover usable bases from the SSC clusters?
Short, high-level explanations of all these build blocks would help tremendously.  The intended inputs and outputs to all steps should be specified.

 4) The MCD-kmeans and MCD-PCA approaches mentioned in p 5 should be moved to the experiments.  This would keep the focus on the proposed approach.

 5) The complement subspace $C^{n_c+1}$ defined in p 4 should be introduced later, when the need for it becomes apparent.  Also, $C^{n_c+1}$ does not roll easily on the tongue.  I suggest to introduce a more obvious mnemonic, say $C^\bot$, $D$, or similar.

 6) In the experiments, clearly state all reasearch questions and provide clear descriptions of all data sets and evaluation metrics.

- Eq. 2: I suggest to use use a different symbol for $\phi^l_i$, to avoid confusion with the feature vector $\phi$.

- p 6: Footnote 4 should be moved next to footnote 3.

- p 6: In the first sentence on p 6, change "i.e., $max_{x,y} ...$" -> "i.e., we divide the vectors elementwise by $max_{x,y} ...$".

- p 6: In the second sentence, what is $|\phi^l|$?  Is it the Euclidean norm?  If so, please use $\|\phi\|_2$ instead.  Otherwise, please explain what $|\cdot|$ means.

- p 8: Sec 3. "The segmentation into candidate concept patches is model-independent and thus, segments are not necessarily meaningful as perceived by the model".  But by locality - which holds in CNNs and (as the authors mention) empirically also for ViTs - superpixels generally map to subsets of latent features.

- p 9: What does "yellow(white)" and "0.5(0.4)" mean?

- p 9: "the the"

- Everywhere: "true-to-the-model" - I suppose the authors refer to "faithfulness".  If so, the latter term should be preferred as it is more common in the XAI literature.

- Why are some of the equations boxed?  The boxes are unnecessary and should be removed.

- Fig 2: It would be useful to draw $\phi^\alpha_{xy}$ as a tube.  Also, please rotate the $x$ and $y$ coordinates, for readability.


*Quality* I have some issues with the message conveyed by the paper.  Specifically:

- p 2: "Allowing for arbitrary multi-dimensional subspaces unfolds the most general deﬁnition of a linear decomposition. Thus, this general approach enables true-to-the-model concepts as it allows to capture any meaningful linear structure within the hidden feature layer (benefit 1)."  I agree with the first sentence - linear subspaces are the most complex linear substructure possible for a linear space - but not with the second.  Why are linear subspaces "true-to-the-model"?  Please, either remove the second sentence (as it doesn't add much), make it relative (e.g., "truer-to-the-model" or "more faithful"), or provide arguments in support for it.  The reason why I am insisting on this is that the "true-to-the-model" slogan is repeated again in the text - but to the best of my knowledge it is not justified: the model may be carrying out non-linear operations during prediction, so faithfulness is not guaranteed.  I think it's important to discuss this point thoroughly, as faithfulness is one of the main selling points of the technique.

- p 4: Correlated to the above:  the authors write "Note, that exploring even more general structures, such as concepts as sub-manifolds in feature space, is an interesting idea. However, these do not allow for a decomposition of the feature vector and hence do not lead to a completeness property, which is central to the deﬁnition of concept relevance maps".  The meaning is unclear to me.  I'd appreciate if the authors could elaborate on what they mean.  A second observation is that this seems to be in conflict with the "true-to-the-model" motto: if the model does reason in terms of manifolds rather than linear subspaces, explanations given in terms of subspaces (that as such satisfy the completeness property) cannot possibly be faithful to the model.  So I'm not sure what benefits completeness yields - especially from the perspective of faithfulness.  My understanding is that they are both useful properties, but quite orthogonal to each other.  They authors should discuss this briefly.

- p 4: "assume without loss of generality that their subspaces are pairwise disjoint."  Why is this w.l.o.g.?

- p 4: "does not take advantage of the spatial proximity of the data" -> "does not rely on ..." (the sentence, as it is, implies a negative connotation)


*Experiments*:

1) The SCD dataset is not described, so I don't know whether this evaluation is in any way useful.

3) Ideally, MCD should be compared to ground-truth explanations.  To this end, the authors could define data sets where the ground-truth explanations are known, for instance, using data sets with known concept annotations and defining synthetic labels based on simple logical combinations of concepts, then testing whether MCD can recover the concepts responsible for the label.  Right now, it is very hard to know whether the MCD explanations are faithful to the concept's reasoning.  This is optional, but it would help immensely.



*Related Work*  The related work is well done.  Two aspects that may be worth discussing are:

1) Automatically extracted concepts - even those featuring nice theoretical properties like completeness - are not necessarily human understandable.  This is a very important aspect.  In fact, assuming concept interpretability is viewed as concept identification - as done in:

 Marconato et al., "GlanceNets: Interpretabile, Leak-proof Concept-based Models", NeurIPS'22.

and implicitly in other works on concept-based explanations - it is very hard to ensure concepts extracted automatically from networks are interpretable.  This is exactly the reason why TCAV, Concept-Bottleneck Models, Concept Whitening, and GlanceNets require a modicum of concept supervision.  The authors should explain what makes concepts extracted by MCD intepretable (regardless of the how the network under examination is structured and learned), and discuss possible limitations of their (unsupervised) approach.

2) In the same vein, it would be worth briefly discussing whether concept supervision could be injected into the explanation procedure to guide the subspace decomposition.

3) The idea of implementing concepts as subspaces has been considered in:

  Zarlenga et al., "Concept embedding models", NeurIPS'22.

It would make sense to discuss the relationship between their (ante-hoc) solution to the proposed (post-hoc) approach.

**Strengths And Weaknesses:**

+ English is good.
+ Contribution generalizes existing approaches in a sensible manner.
+ Contribution is of interest to the XAI/CBM community.
+ Related work is well done.

- The structure of the text should be improved.
- Some comments are not entirely motivated.

---

> ### Author Response · Authors · 2023-03-27
> **Improved readability, resolved conflict between faithfulness and completeness, and relation to disentanglement learning**
>
> **Structure and clarity (writing)** We are very grateful for the reviewer’s very detailed and constructive suggestions, which helped to greatly improve the readability of our paper. As requested, we rephrased the introduction and now use a simple example to illustrate the various concept definitions and thereby ensure accessibility to a broad audience. In line with the reviewer, we have deferred technical details to a brief description of the MCD algorithm at the beginning of section 2, which now outlines the expected inputs and outcomes of all steps. We further followed the reviewer's suggestions and moved algorithmic details from section 2.1 (concept definition), to section 2.2. We added more details to section 2.2., describing the intuition behind SSC and our choice of PCA, and moved the description of additional MCD flavors to a dedicated new section 2.4.
>
> **Conflict between faithfulness and completeness (quality)** We thank the reviewer for pointing us to the need to delimit benefit 1 and the alleged conflict between completeness and faithfulness, introduced by the cited sentence in section 2.1, that can be resolved easily. While it is true, that we cannot completely decompose feature vectors into non-linear concepts, such as sub-manifolds, there is no point in differentiating such non-linear structures that cannot be separated by the linear classifier for the last hidden feature layer we specialize to for the definition of concept relevance heatmaps. We account for this with two changes: First, we made the statement on benefit 1 relative. Second, we deepened the discussion around non-linear concept structures, and when linear subspaces are faithful or only represent an improvement over more constraint concept definitions in section 2.1.
>
> **Concept supervision and relation to disentanglement learning (related work)** Also here, we are very grateful for the reviewer’s remarks, as it helped to expand towards a so far only sparsely covered direction of related research but, more importantly, helped us to rethink the central differences between these approaches and the proposed methods.
> We have extended the discussion on inherently interpretable models by design, in particular Concept Bottleneck Models (CBMs), in the related works section. To summarize, there is a crucial difference in enforcing concept interpretability between ante-hoc and post-hoc approaches. It makes sense to regularize/supervise training of CBMs for high concept interpretability but enforcing it for post-hoc explanations obfuscates the explanation and renders the model more interpretable than it actually is. In the end, MCD could also reveal that the model has not learned any structures that align with human concepts.
>
> **Experiments** (1) We previously described the dataset, model, and choice of feature layer for the SDC experiment in 4.2.1 and 4.2.2 in the beginning of section 4.2. We now reiterate them in both subsections for better readability.  (2) SDC already provides a relative comparison between the faithfulness of different concept-based explanation methods. The requested experiment to test absolute faithfulness requires a well-controlled setting, where no confounding factors can obfuscate the ground-truth, e.g. synthetic data. We believe that the construction of a synthetic dataset, that is complex enough for the model to learn non-trivial structures in feature space, is beyond the scope of this work.
>
>  **Broader impact concerns** We added a discussion on the identifiability of MCD concepts with known human concepts to the conclusion. MCD does not provide guarantees for identifiability, because MCD prioritizes faithfulness and therefore does not enforce interpretability.
>
> **Small changes** We thank the reviewer for pointing us at unclear formulations etc. on pages 4,6,8,9 and in Fig. 2 and Eq. 2. We have changed these. In particular, we have replaced the term ‘true-to-the-modelness' with ‘faithfulness’, which is indeed the standard term for this property, and renamed the orthogonal complement. Further, we removed the w.l.o.g. statement on page 4 and moved the footnote with the algorithmic details on how to fulfill pairwise disjointness of concept subspaces to section 2.3, where it becomes relevant.

---

### Review · Reviewer_MjCK · 2023-03-17

**Summary Of Contributions:**

This paper proposes a method to extract "concepts" in deep image classification models as subspaces in latent space.
In previous research, "concepts" are typically defined as vectors in latent space.
The idea in this paper can be thus interpreted as a generalization of existing "concepts".
The authors pointed out that existing sparse subspace clustering techniques can be used to estimate these "concept" subspaces.
The authors also proposed a method to quantify the contribution of each "concept" in image classification using the basis vectors of each subspace, and to extract prototypical data for each "concept".
Furthermore, the authors enabled to visualize the contribution of each "concept" in the image through heatmaps.
The generalization of the notion of "concepts", the extraction of prototypical data, and the visualization through heatmaps are useful tools for users to understand deep image classification models.
The authors also conducted experiments using ResNet and Swtin Transformer, demonstrating that the proposed method is more effective in extracting "concepts", and that there are differences in the found "concepts" between models.

**Audience:**

Yes

**Broader Impact Concerns:**

There is no ethical concern.

**Claims And Evidence:**

Yes

**Requested Changes:**

#### Request 1.
I recommend the authors to enlarge images in Figures 3 and 5 as thy are too small to inspect.

#### Request 2.
I recommend the authors to move some (not necessarily all) of Figures 6, 7, and 8 to the main body of the paper.
They are essential ingredients and should be presented in the main body.

**Strengths And Weaknesses:**

### Strong aspects

#### Strength 1: Generalization of "concepts"
The major contribution of this study lies in the definition of "concepts" as subspaces in latent space.
In previous research, "concepts" are typically defined as vectors in latent space.
However, it would not be natural to consider that complex "concepts" can be represented as a single vector in high-dimensional latent space.
On the other hand, subspaces can more naturally represent richer and more complex "concepts" by adjusting their dimensions appropriately.
This perspective would be an important step forward in the study of "concepts".

#### Strength 2: Benefits of subspace
The strengths of this study is not only the generalization of the definition of "concepts", but also the extraction of prototypical data and the visualization through heatmaps, which will further help users to understand deep image classification models.
Furthermore, the extraction of these prototypical data and the visualization through heatmaps are naturally realized through the basis expansion of the latent vectors as well as the weight coefficients of using the basis vectors of the subspaces.
This suggests that defining "concepts" as subspaces is also useful as a model interpretation tool.


### Weak aspects
I did not find any crucial weakness.
As minor suggestions, I recommend the authors to enlarge images in Figures 3 and 5 as thy are too small to inspect.
I also recommend the authors to move some (not necessarily all) of Figures 6, 7, and 8 to the main body of the paper.
They are essential ingredients and should be presented in the main body.

---

> ### Author Response · Authors · 2023-03-27
> **Additional and enlarged figures**
>
> We thank the reviewer for the positive assessment of our work. We also thank the reviewer for the request to enlarge Figures 3 and 4, which we followed in the revised version of our manuscript. Further, we agree that the results in Figures 6-8 are essential and moved Figure 7 to the main body to support our findings in section 4.2.1.

---

### Review · Reviewer_2vJn · 2023-03-20

**Summary Of Contributions:**

This paper proposes a framework  for concept discovery based on hidden representation of neural networks. Different from existing approaches, authors proposed a multi-dimensional linear subspaces without restricting to single directions or enforcing orthogonality between subspaces with complete relations. Empirical results show better performance.

**Audience:**

Yes

**Claims And Evidence:**

No

**Requested Changes:**

clear and sound justification on why multi-dimension and non-orthogonality are desired.

**Strengths And Weaknesses:**

Strengths

- Empirical results seem better

Weakness

- Presentation is not very clear. Many related concepts are not fully explained, and motivations are not clearly written. Hence, one cannot fully judge the motivation or understand the advantages of the proposed approach.
- Overall, I find the paper lacks sound justification, and mostly are hand wavy explanations. (although I recognize that maybe this is standard in XAI field due to lack of theoretical understanding and very subjective.)


Comments:
- XAI is not explained in abstract.
- "directly identify concepts with a neural directions": is it one direction or 3?
- losing orthogonality seems to lead to overlap of concepts. I'm not sure arbitrary orientation is a desired property for concise explanation, since they likely contain redundant information. More discussion (esp. theoretical justification) here would be helpful.

Section 4.2.1 studies this aspect in more details. However, it is not fully clear to me why MCD-SSC is better than MCD-SSC-ortho. "the greedy way these orthogonal subspaces are constructed" - also very informal. If one purposely control more concepts to be desired (maybe via some hyperparameters tuning), would it result in more fine-grained concepts for MCD-SSC-ortho?

- "Multi-dimensionality ensures concise explanations": although this is discussed in introduction, I don't find this argument clear. Later section also lacks a clear mathematical justifications on why higher dimension is better than one-dimension. In addition, what would a good way to determine the dimension size?
- Some hyperparameter choices are not well motivated, via ablation study for example. For example, number of concepts.
- How does the propose method differ from many disentangled representation learning or prototype learning?

---

> ### Author Response · Authors · 2023-03-27
> **Advantages of multi-dimensional and non-orthogonal concepts**
>
> **Overall presentation** We thank the reviewer for suggesting to rework the presentation of our work. We improved the overall structure and presentation, particularly within the introduction and the methods section. We amended the introduction and clarified crucial passages, also including an illustrative example, to make our motivation for concepts as multi-dimensional subspaces clear. In the methods section, we added an overview of the MCD algorithm, and explained our methodological choices like SSC and PCA more thoroughly.  Further, we introduced a dedicated section explaining the alternative MCD flavors. In particular, we rephrased the description of MCD-SSC-ortho, to clarify how it constructs orthogonal subspaces.  Further, we now introduce the acronym XAI in the abstract and improved the unclear formulation in the introduction.
>
> **Advantages of multi-dimensional concepts*** Multi-dim. concepts are the most general linear  structure a that can be exploited by a model which builds on a linear classification head. There is no theoretical justification for assuming that concepts form a less general structure, i.e. single directions. Therefore, multi-dim. subspaces are the most faithful linear concept definition, as they do not make any assumptions that are not justified by the model architecture or training. The conciseness of the explanation, which means that it comprises only a small number of concepts, follows directly from the multi-dimensional structure of concepts. Contrary, to obtain a faithful coverage of feature space many more one-dimensional concepts are required. We experimentally observe this phenomenon in Tab. 2. Here, approximately 30x more one-dim. concepts are necessary to reach the same faithfulness to the model reasoning. In section 2.2, we propose a way to infer the concept subspace dimensions from a set of clustered feature vectors via PCA.
>
> **Advantages of non-orthogonality** We thank the reviewer for bringing up this important point. As faithfulness is the central design principle of MCD, we include no restrictions in our concept definition other than requiring concepts to form linear structures. Generally, we are not aware of any direct mechanism, that enforces orthogonality in hidden feature space during model training. We added a statement emphasizing this crucial ingrediency to the introduction. Concerning conciseness, the union of n arbitrarily – but not parallelly – oriented concept subspaces C^l has the same dimensionality as the union of the same n concept subspaces after rotating them such that they are orthogonal to each other. Phrased differently, by re-introducing orthogonality, we cannot decrease the number of concepts, i.e., gain conciseness, when the subspace of the feature space to be explained remains fixed. We emphasize that it is possible to lose faithfulness by this restriction, as the actual concepts on which model relies on might not be orthogonal. From a practical perspective, some concepts might naturally lie close in feature space, and thus appear to have overlap in concept activation maps.
>
> Regarding the empirical evidence for the advantage of not requiring orthogonality, in Section 4.2.1, we rephrased the discussion around the impact of concept orthogonality on faithfulness in the SCD experiment.  Further, we modified section 4.2.2 to clarify the trade-off between conciseness, distance between, and dimensionality of concepts.
>
> **Relation to disentanglement and prototype learning** First, disentanglement and prototype learning train new inherently interpretable concept models from scratch, whereas MCD is an interpretable reformulation of an existing non-concept model. MCD is applied post-training and can be leverage to understand existing and already trained models. In particular, the objective of disentanglement learning is to recover the concepts hidden in the underlying data generation process, while MCD discovers concepts in a hidden feature layer, i.e. those learned by the model.  We have extended the related work section, which now includes a perspective on disentanglement learning and prototype models.
>
> **Hyperparameter choice** As the reviewer pointed out, the most important hyperparameter to MCD-SSC is the number of concepts. Our work novelly offers a meaningful way to fix this hyperparameter by choosing the degree of completeness, to which the discovered concepts should cover the feature space. We added a clarifying remark to section 2.3. The remaining hyperparameter is the coefficient gamma in the SSC objective, which balances sparsity vs robustness. We already describe that the discovered concepts are insensitive to this in appendix A.
>
> In summary, the modified draft now offers clear arguments and evidence to justify all claims.

---

### Author Response · Authors · 2023-03-27
**Brief overview and revised manuscript**

We appreciate the positive comments by the three reviewers, all three do not voice any major technical or methodological concerns. We especially welcome the constructive suggestions made by the reviewers to improve the presentation of our work, which we have fully taken into account in our revised manuscript.

**Reviewer FPuj** gives a generally positive assessment of our work. He/she approves that our work is relevant and appropriate contribution for TMLR. We want to thank the reviewer again for his/her in-depth high-quality review, which helped to immensely improve the readability and clarity of our manuscript.

**Reviewer MjCK17** strongly agrees with the contribution of our manuscript and emphasizes the significance of our findings for the field of concept-based explanations. We have incorporated his/her suggestions to improve the presentation of our results.

**Reviewer 2vJn20** appreciates the empirical results that demonstrate the superiority of MCD–SSC over other methods. We thank the reviewer for his/her review. In particular, the request for clarification of the disadvantages of concept orthogonality, for which we provide further arguments in the revised manuscript, helped to further justify the superiority of MCD.

We posted a point-by-point responses to each of the comments of the reviewers. In the revised manuscript, we have addressed all specific suggestions and calls for clarification. Revisions in the text are highlighted in blue. We hope that the revised manuscript will be deemed appropriate for final publication in Transactions on Machine Learning Research

---

### Decision · Action_Editors · 2023-05-03

**Recommendation:** Accept as is

**Comment:**

The authors propose a new post-hoc explainable AI method, which leverages multi-dimensional linear subspaces without orthogonality restrictions to discover hidden concepts. Empirical results demonstrate the potential of the proposed method.

The reviewers agree about the contribution of the work to the problem of concept-based explanation. The presentation issues of the first draft were fixed during the revision. While some reviewers have concerns about the theoretical justifications of the proposed method, the empirical results are convincing enough to validate the proposed method and support the claim on the possible advantages of multi-dimensional concepts without orthogonality restrictions. All reviewers lean acceptance after the revision.



**Audience:**

Yes

**Claims And Evidence:**

Yes